# Bovine host genome acts on rumen microbiome function linked to methane emissions

Marina Martínez-Álvaro [1], Marc D. Auffret[2], Carol-Anne Duthie[1], Richard J. Dewhurst [1], Matthew A. Cleveland[3], Mick Watson [4] & Rainer Roehe [1]✉

Our study provides substantial evidence that the host genome affects the comprehensive function of the microbiome in the rumen of bovines. Of 1,107/225/1,141 rumen microbial genera/metagenome assembled uncultured genomes (RUGs)/genes identified from whole metagenomics sequencing, 194/14/337 had significant host genomic effects (heritabilities ranging from 0.13 to 0.61), revealing that substantial variation of the microbiome is under host genomic control. We found 29/22/115 microbial genera/RUGs/genes host-genomically correlated (|0.59| to |0.93|) with emissions of the potent greenhouse gas methane ($CH_4$), highlighting the strength of a common host genomic control of specific microbial processes and $CH_4$. Only one of these microbial genes was directly involved in methanogenesis (*cofG*), whereas others were involved in providing substrates for archaea (e.g. *bcd* and *pccB*), important microbial interspecies communication mechanisms (*ABC.PE.P*), host-microbiome interaction (*TSTA3*) and genetic information processes (*RP-L35*). In our population, selection based on abundances of the 30 most informative microbial genes provided a mitigation potential of 17% of mean $CH_4$ emissions per generation, which is higher than for selection based on measured $CH_4$ using respiration chambers (13%), indicating the high potential of microbiome-driven breeding to cumulatively reduce $CH_4$ emissions and mitigate climate change.

[1] Scotland's Rural College, Edinburgh, UK. [2] Agrifirm, Drongen, Belgium. [3] Genus plc, DeForest, WI, USA. [4] The Roslin Institute and the Royal (Dick) School of Veterinary Studies, University of Edinburgh, Edinburgh, UK. ✉email: Rainer.Roehe@sruc.ac.uk

Ruminants harbour a unique symbiotic gut microbial population that transforms indigestible fibrous feed into high-quality products such as meat and milk for human consumption. Moreover, ruminant livestock is vital to meet global food security and contribute to poverty[1] reduction in an increasing world population[2]. Yet to be solved is the negative environmental impact, as dairy and beef cattle account for 9.5% of all anthropogenic greenhouse gas (GHG) emissions[3]. Of those, ruminal microbial fermentation represents 40–50%; in particular, due to the highly potent GHG methane ($CH_4$)[4]. Additionally, $CH_4$ emissions imply a considerable energy loss to the animal, ranging from 2 to 12% of gross energy intake[5]. Therefore, decreasing $CH_4$ emissions is acknowledged to contribute to the mitigation of climate change and optimize the economic efficiency of cattle production[6]. Ruminal methanogenesis is a complex process dependent on the cooperation of taxonomic communities with different metabolic activities[7–10]. A diverse community of bacteria, ciliate protozoa, and anaerobic fungi[11] convert complex diet carbohydrates, proteins, and lipids into volatile fatty acids, lactate, microbial proteins, and vitamins while releasing $CO_2$, $H_2$, and other compounds. Four orders of ruminal methanogenic archaea use electrons derived from $H_2$, formate or methyl compounds to reduce carbon dioxide into $CH_4$ to obtain energy for growth[12]. Dietary interventions designed to alter the microbiome for $CH_4$ mitigation (e.g. protozoa defaunation[13,14], seaweed[15], and 3-NOP[16] additives) have often failed in the long term due to microbiota adaptation to the new environment[17] or are associated with increased production costs. In contrast, the genomic selection that targets the part of the host genome modulating microbiome composition related to low $CH_4$-emitting cattle opens up the opportunity to provide a cost-effective permanent solution to reduce $CH_4$ emissions from ruminants.

There is increasing evidence of a host genetic impact on the composition of the microbiota in the rumen of bovines[7,18–25], monogastric livestock[26,27], and humans[28–33]. These previous metagenomic studies were based on microbiota profiles mostly identified using sequence polymorphisms of the 16S rRNA gene and therefore did not consider the functional versatility of unique rumen microbial species, nor the ability of some microbial organisms to integrate foreign DNA from other microbial organisms into their DNA. Novel microbial species in the rumen have recently been identified using metagenome-assembled genomes generated from whole metagenomic sequence data of microbial DNA from rumen samples[8,34], but how their abundances are shaped by host genomics is still unknown. There is also a lack of knowledge of the host genomic associations with the abundances of functional microbial KEGG genes which were found to predict methane emissions on the phenotypic level with a prediction ability of $r^2 = 0.81$[7]. Moreover, a microbiome-driven selection strategy based on this functional information of microbial genes in animal breeding and a study of its effect on the response to selection is to the best of our knowledge not available, since previous research emphasized host genetic effects on the taxonomical composition of the microbiome[22,35,36]. To identify the functionality of the rumen microbiome directly, we applied genome-resolved metagenomics to generate abundances of the microbial KEGG genes based on whole metagenomic sequencing. Based on these data, we carried out comprehensive research that elucidates how host genomics influences complex functions of ruminal microbes (determined by their microbial genes) genomically correlated to $CH_4$ emissions. Furthermore, we developed a microbiome-driven selection strategy showing how this information can best be included in cattle breeding to directly change the rumen microbiome function and reduce these emissions.

We identified the host genomic (using single-nucleotide polymorphism (SNP) information) influence on an extensively characterized microbiome in relation to $CH_4$, using whole metagenome sequencing of rumen microbial DNA samples from a bovine population designed for a powerful host genomic analysis[37–39] with high standardization of diets and other husbandry effects. We characterized the core ruminal microbiome by identifying 1,108 cultured microbial genera by mapping our sequences to the Hungate1000 reference genome collection[40] and RefSeq[41] databases (Supplementary Data 1); 225 ruminal uncultured genomes (RUGs) by de novo metagenome-assembly of genomes[34], 34 of them classified at strain level (Supplementary Data 2), and 1142 functional microbial genes (Supplementary Data 3); present in all ($n = 359$) and for RUGs in >200 of our animals. Our specific hypothesis is that the host genome influences the abundance of not only functional microbial genes involved in metabolism, but also in interspecies communication, host–microbiome interactions, and genetic information processing. These functions may play a key integrating role in achieving a ruminal balance where fermentation of feed into essential nutrients utilized by the host is optimized and substrates utilized by methanogenesis e.g. $H_2$ excess are minimized. Our comprehensive description of ruminal microbiome functionalities includes the abundances of 34 microbial genes carried by methanogenic archaea directly implicated in $CH_4$ metabolism; 511 involved in other metabolic pathways of bacteria, archaea, ciliate protozoa or fungi, indirectly influencing methanogenesis by minimizing required substrates through non-methanogenic routes that yield beneficial nutrients for ruminants[42] (e.g. acetogenesis, propionogenesis[13,43–45]), or generating methanogen-inhibitor metabolites[46–48]; 207 in microbial communication processes and host-microbiome interaction (e.g. ABC transporters of different metabolites or fucose sensing) carried by fungi, bacteria and archaea[49–51], of importance because the synthesis of $CH_4$ in cooperation with other main metabolic routes in the rumen[52–54] are syntrophic processes amongst microbial communities[55]; 330 involved in genetic information processes (e.g. ribosomal biosynthesis) related to microbial growth[56]; and 60 at present not functionally characterized. For each of these 2475 functional and taxonomic characteristics of the rumen microbiome, the host genomic determination and correlation with $CH_4$ emissions were analysed. After stringent adjustment for multiple testing, we demonstrate that our hypothesis of a common host genomic control is valid by discovering heritabilities ($h^2$) of microbial profiles and host-genomic correlations with $CH_4$ emissions ($r_{gCH4}$) significantly deviating from zero, which shows the effectiveness of this strategy. Our results are obtained in bovines, but also provide an indication of potential host genomic effects on functional microbial genes and their biological processes in other species.

Besides providing a better understanding of the complex host genomic effects on the rumen microbiome function, this research provides the basis for a cost-effective microbiome-driven breeding strategy to mitigate $CH_4$ emissions from cattle without measuring it directly, which is necessary considering the cost-prohibitive limitations of measuring individual animal $CH_4$ emissions.

## Results

**Bovine host genomics affected $CH_4$ emissions produced by ruminal archaea.** $CH_4$ emissions[57] were accurately measured from individual beef cattle ($n = 285$) using the gold-standard method of respiration chambers. Animals within the same breed or diet expressed high phenotypic variability in $CH_4$ emissions with coefficients of variation from 16.3 to 28.5% (Supplementary Fig. 1a, b). Genomic $h^2$ of $CH_4$ emissions revealed that 33%

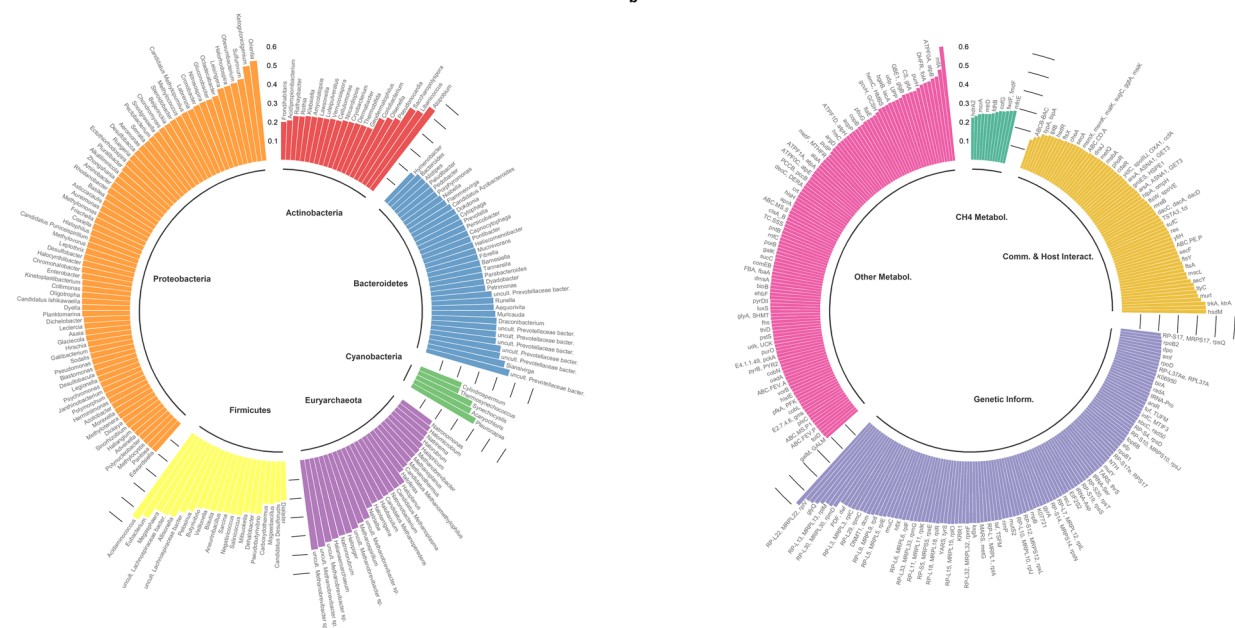

**Fig. 1 Genomic heritability ($h^2$) estimates of additive log-ratio-transformed abundances of microbial taxa and their genes in the rumen of 359 bovines.** Bars show the $h^2$ values of 194/14/337 rumen microbial genera/uncultured genomes (RUGs)/genes tested exhibiting non-zero $h^2$ estimates and significant host genomic effects (based on Bayes Factor >3 and Deviance Information Criterion difference between models with or without host genomic effects ≤−20). **a** Cultured microbial genera and RUGs classified within the phylum. **b** Microbial genes grouped by microbial biological processes: microbial communication and host-microbiome interaction (Comm. & Host Interact.), genetic information processes (Genetic Inform.), metabolism other than methane (Other Metabol.), and methane metabolism ($CH_4$ Metabol.). Source data is in Supplementary Data 4–6.

(Bayes Factor for genomic effects (BF) = 5.91) of this phenotypic variation was explained by host genome variation, which is consistent with other studies[58–60]. The $h^2$ obtained for $CH_4$ emissions is at the level of other traits for which substantial gains due to breeding are achieved, such as growth rate[61] and milk yield[62]. In addition, there was large genomic variation for $CH_4$ emissions as deviation from the mean ranged from −2.67 to 3.51 g/kg of dry matter intake (DMI) with no difference between breeds ($P > 0.16$), which suggests that bovines have most likely not been indirectly selected for $CH_4$ emissions as a result of a lack of genetic correlation to those traits under selection.

**Host genomics affects the ruminal microbiome composition.** We next investigated the proportion of the ruminal microbiome variation at taxonomic and functional levels explained by the host genomic variation among individuals, by estimating $h^2$ of the ruminal abundances of 1107 genera, 225 RUGs, and 1141 microbial genes. Our results demonstrate significant host genomic effects with $h^2$ in a range between 0.13 and 0.61 for the abundances of 194 microbial genera, 14 RUGs, and 337 microbial genes representing cumulatively 58.4, 5.63, and 27.2%, respectively, of the total relative abundance (RA) (Fig. 1 and Supplementary Data 4–6). Among the 194 genera, 20 were highly heritable ($h^2 > 0.40$), which belonged exclusively to bacteria (e.g. Firmicutes *Acidaminococcus* (RA = 0.3%), $h^2 = 0.54$, BF = $8.82 \times 10^{+5}$) and archaea (e.g. hydrogenotrophic methanogen *Methanospirillum*[63] (RA = 0.0005%, $h^2 = 0.40$, BF =; $1.04 \times 10^{+2}$). Host genome also shaped the abundance of the hydrogenotrophic/methylotrophic[55] methanogen *Candidatus Methanoplasma* (RA = 0.002%, $h^2 = 0.32$, BF = $6.31 \times 10^{+2}$), and to a lesser extent the abundance of ubiquitous *Methanobrevibacter* (RA = 5.02%, $h^2 = 0.24$, BF = 9.10)—which is coherent with estimates from other studies[18–20]—*Candidatus Methanomethylophilus* (RA = 0.05%, BF = 4.18), *Methanothermus* (RA = 0.002%, $h^2 = $

0.25, BF = $1.26 \times 10^{+1}$) and *Methanoplanus* (RA = 0.0008%, $h^2 = 0.24$, BF = 9.56). Reinforcing the evidence of a host-genomic component in the abundance of methanogenic archaea, 5 RUGs annotated as uncultured *Methanobrevibacter* strains (RA > 0.27%) demonstrated moderate to high $h^2$ estimates (0.39–0.48, BF from 3.5 to $4.65 \times 10^{+1}$), indicating that more specific classification using RUGs provides the opportunity to find highly heritable strains. The most abundant complex carbohydrates degraders in the rumen—*Eubacterium* (RA = 1.02%), *Prevotella* (RA = 39.2%), *Butyrivibrio* (RA = 2.54%), *Bacteroides* (RA = 1.39%) and *Pseudibutyrivibrio* (RA = 0.54%)—were highly ($h^2 = 0.51$ for *Eubacterium*, BF = $9.72 \times 10^{+3}$) or moderately ($h^2 = 0.23–0.33$ for the others, BF from 7.42 to $9.73 \times 10^{+1}$) heritable; with 7 highly abundant RUGs (RA > 0.25%) classified as uncultured *Prevotellaceae bacterium* having $h^2$ from 0.32 to 0.45 (BF from 7.48 to $1.67 \times 10^{+2}$). These results support the concepts of a "core heritable microbiome"[21,64] and stability over time of certain microbial genera abundance such as *Prevotella*[65]. None of the fungi and protist genera, which are considered to be non-essential for rumen function and highly variable within different host species[66], were highly heritable.

We elucidated that the specific functional capacity of the ruminal microbiome is heritable by estimating the $h^2$ of a comprehensive set of microbial genes, of which 33 were highly ($h^2 > 0.4$), and 304 were moderately ($0.2 < h^2 < 0.4$) heritable. These microbial genes are involved in a wide variety of metabolic functions (Fig. 1b and Supplementary Data 6), e.g. synthesis of microbial proteins or volatile fatty acids, suggesting that the host genome influences the growth of microbes responsible for the release of nutrients during microbial fermentation[67,68]. Among 34 microbial genes involved in the $CH_4$ metabolism pathway, 13 showed moderate $h^2$ of 0.22–0.29 (BF from 3.96 to $7.82 \times 10^{+1}$), e.g. *mcrA*, *mcrB*, *mtrD*, *mtrE*, and *cofG*. Ribosomal biosynthesis was revealed to be under strong host-genomic

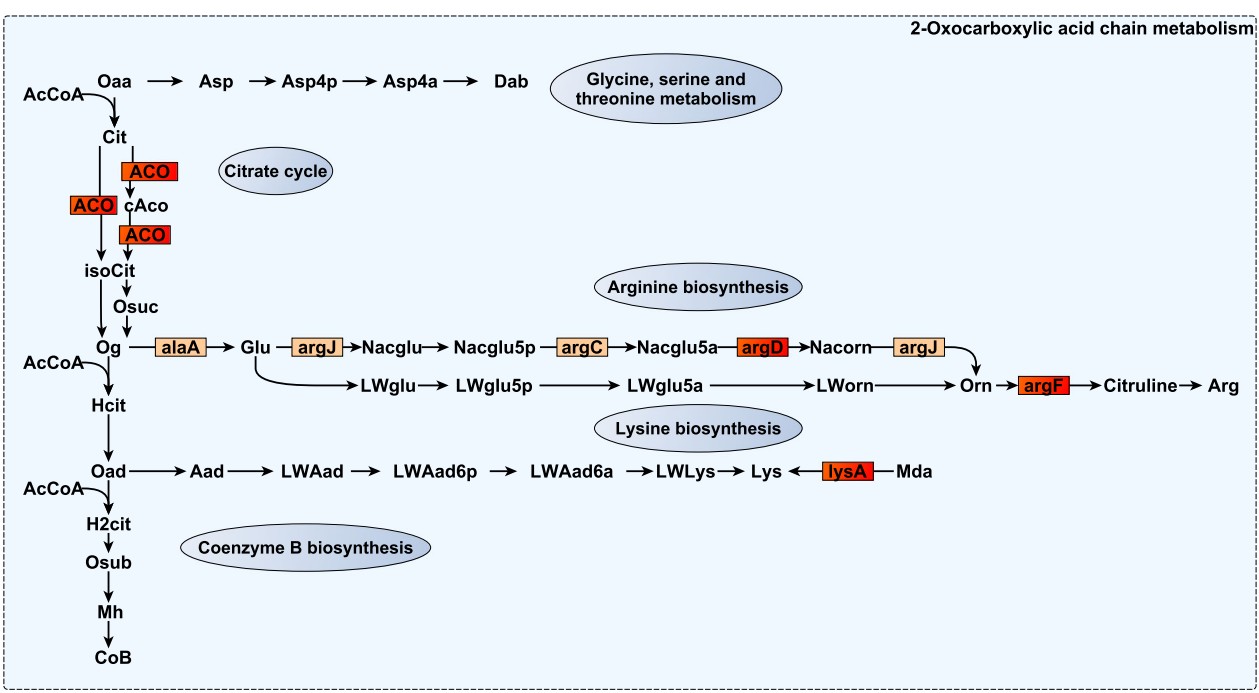

**Fig. 2 Reaction scheme of 2-oxocarboxylic acid metabolism and glycine, serine, threonine, arginine, lysine, and Coenzyme B biosynthesis in which additive log-ratio transformed microbial gene abundances strongly host-genomically correlated with methane emissions ($r_{gCH4}$) are involved.** Small rectangles symbolize proteins encoded by the microbial genes. Microbial genes are highlighted in red when their $r_{gCH4}$ estimates range between −0.74 and −0.93 and show a probability of being different from 0 ($P_0$) ≥ 0.95, and in orange when they range between |0.55| and |0.77| and $P_0$ ≥ 0.85. Source data is in Supplementary Data 9. Compounds are denoted by their short names. Full names of compounds and microbial genes are given in Supplementary Data 17.

control with 56 heritable microbial genes, representing a cumulative RA of 6.57%, including 9 highly heritable genes ($h^2 = 0.40$–0.53, BF = $1.62 \times 10^{+2}$ to $1.18 \times 10^{+8}$) synthesizing the large ribosomal subunit. Intracellular ribosomal biosynthesis reflects the growth rate of microbial organisms, given that ribosomes can account for up to 40% of their cellular dry mass[56], and cell fitness and optimal growth are tightly coupled to efficient protein synthesis[69]. Demonstrating that differences among animals in complex microbiome functions are partly due to host genomic variation opens up opportunities to consider a new source of genetic variation not only in ruminants but also in humans, where the $h^2$ of microbial gene abundances was estimated to be even larger (0.65–0.91)[70].

**Ruminal microbial mechanisms related to CH$_4$ emissions are influenced by host genomics.** The existence of a common host genomic influence on CH$_4$ emissions and the rumen microbiome was evaluated by estimating host-genomic correlations between CH$_4$ emissions and each microbial genus/RUG/gene abundance ($r_{gCH4}$). Based on the probability of $r_{gCH4}$ being different from 0 ($P_0$) ≥ 0.95, our study revealed 29 microbial genera, 22 RUGs, and 115 functional microbial genes strongly host-genomically correlated with CH$_4$ emissions ($r_{gCH4}$ from |0.59| to |0.93|, Supplementary Data 7–9). Among the significant microbial communities, most were bacteria (22 genera/17 RUGs) belonging to Bacteroidetes (5/14), Firmicutes (6/2), and Proteobacteria (9/1) phyla. Most microbial genes with strong $r_{gCH4}$ were not directly involved in CH$_4$ metabolism pathways, but rather mechanisms indirectly affecting CH$_4$ production most likely by limiting substrates for methanogenesis[10,71], inhibiting methanogens, playing a role in coordinating actions among microbial communities and the host or leading microbial genetic processes. Only H$_2$-oxidizing *Methanoregula* (RA = 0.003%) with unknown activity in rumen[49] and the

microbial gene *cofG* involved in F$_{420}$ coenzyme biosynthesis[72,73] resulted in significant negative $r_{gCH4}$ (−0.82 and −0.71, $P_0$ ≥ 0.95), suggesting that these are abundant under ruminal conditions unfavourable for high CH$_4$ producing methanogens. Four uncultured *Methanobrevibacter* strains showed negative $r_{gCH4}$ (<−0.72, $P_0$ ≥ 0.95) and one was positive (0.91, $P_0$ = 0.99), indicating that the relationship between the abundance of *Methanobrevibacter* and CH$_4$ emissions is complex as different strains may have functional versatility. We hypothesize that some *Methanobrevibacter* sp. can produce CH$_4$ even under a challenging ruminal environment (e.g. low pH value), however, at a substantially lower level than those adapted to more favourable conditions. To visualize which microbial genus/gene abundances in the rumen are influenced by a common host genomic background, we constructed a co-abundance network based on Pearson correlations among deregressed host genomic effects for each microbial genus/RUG/gene (Supplementary Fig. 2 and Supplementary Data 10). This approach revealed co-abundance clusters of bacterial and fungal genera[74] with strong $r_{gCH4}$ and methanogenic archaea, e.g. fungal *Metschnikowia* ($r_{gCH4} = 0.77$, $P_0 = 0.96$) and archaeal *Methanosarcina* (cluster 9 in Supplementary Fig. 2); and of microbial genes not directly involved in CH$_4$ metabolism but with strong $r_{gCH4}$ (e.g. *RP-L6*, $r_{gCH4}$ 0.71, $P_0 = 0.96$) and those involved directly in CH$_4$ metabolism (e.g. *fbaA*, cluster 1 in Supplementary Fig. 2). The most important host-genomically affected ruminal microbial mechanisms associated with CH$_4$ production (based on $r_{gCH4}$) are as follows:

*Microbial metabolism.* An extensive group of microbial genes involved in amino acid metabolic and transport pathways displayed negative $r_{gCH4}$. Part of this group of microbial genes was involved in the biosynthesis of arginine[75] and branched-chain amino acids[76,77] via oxocarboxylic acid metabolism (*argF*, a*rgD* and *ilvA* with $r_{gCH4} = -0.84$ to −0.88, $P_0 ≥ 0.96$; and *argJ*,

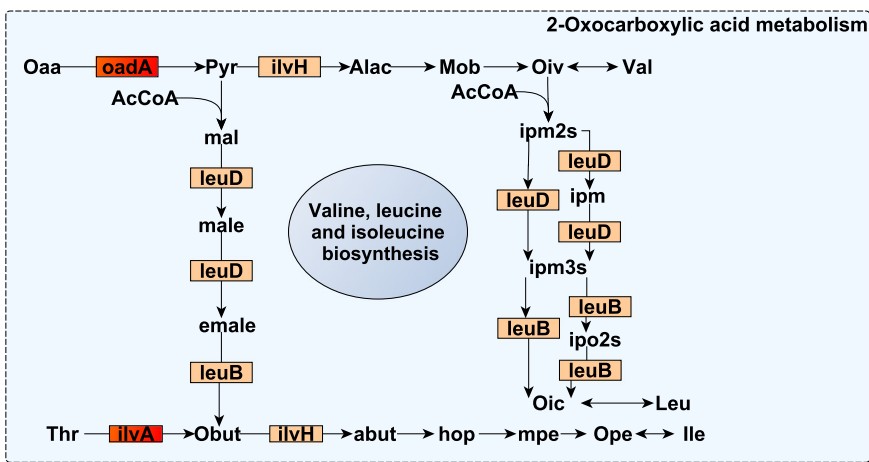

**Fig. 3 Reaction scheme of 2-oxocarboxylic acid metabolism and branched-chain amino acid biosynthesis, in which additive log-ratio transformed microbial gene abundances strongly host genomically correlated with methane emissions ($r_{gCH4}$) are involved.** Small rectangles symbolize proteins encoded by the microbial genes. Microbial genes are highlighted in red when their $r_{gCH4}$ estimates range between −0.74 and −0.93 and show a probability of being different from 0 ($P_0$) ≥ 0.95, and in orange when they range between |0.55| and |0.77| and $P_0$ ≥ 0.85. Source data is in Supplementary Data 9. Compounds are denoted by their short names. Full names of compounds and microbial genes are given in Supplementary Data 17.

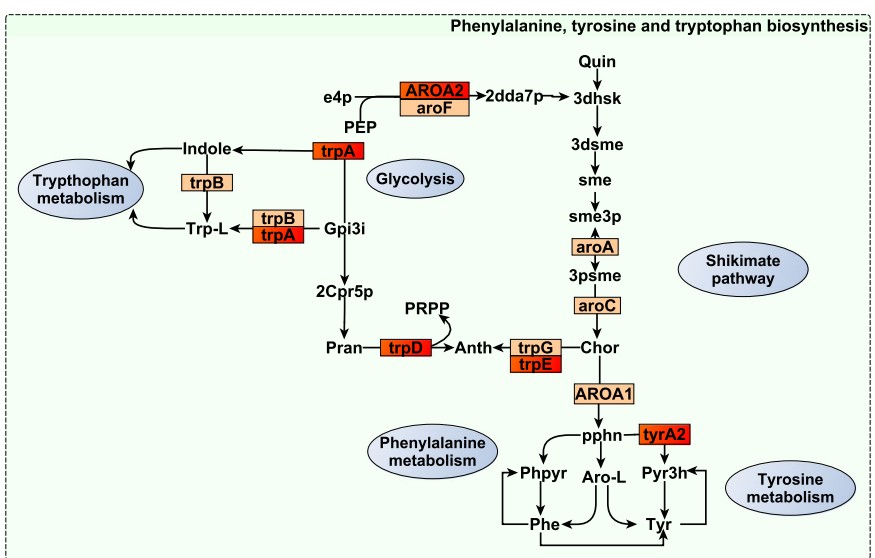

**Fig. 4 Reaction scheme of phenylalanine, tyrosine and tryptophan biosynthesis in which additive log-ratio transformed microbial gene abundances strongly host genomically correlated with methane emissions ($r_{gCH4}$) are involved.** Small rectangles symbolize proteins encoded by the microbial genes. Microbial genes are highlighted in red when their $r_{gCH4}$ estimates range between −0.74 and −0.93 and show a probability of being different from 0 ($P_0$) ≥ 0.95, and in orange when they range between |0.55| and |0.77| and $P_0$ ≥ 0.85. Compounds are denoted by their short names. Source data is in Supplementary Data 9. Full names of compounds and microbial genes are given in Supplementary Data 17.

*argC*, *alaA*, *ilvH*, *leuB*, and *leuD* with $r_{gCH4}$ = −0.55 to −0.77 at lower evidence $P_0$ ≥ 0.85, Figs. 2 and 3). *Aconitate hydratase (ACO)* catalysing the isomerization of citrate to isocitrate in the early stage of the oxocarboxylic chain extension, and *bcd* and *pccB* degrading branched-chain amino acids into branched-chain volatile fatty acids, which have an inhibitory effect on methanogens[46], also expressed negative $r_{gCH4}$ = −0.76 to −0.90 ($P_0$ ≥ 0.95). We also estimated negative $r_{gCH4}$ for microbial genes coding ABC transporters of polar and branched-chain amino acids (*ABC.PA.A*, *ABC.PA.S*, *livH*, *livG*, and *livK* $r_{gCH4}$ = −0.83 and −0.93, $P_0$ ≥ 0.95). Another group of microbial genes was related to the metabolism of aromatic amino acids tryptophan, tyrosine, and phenylalanine (*AROA2*, *trpA*, *trpD*, *trpE*, *tyrA2* and *paaH* with $r_{gCH4}$ = −0.74 to −0.87, $P_0$ ≥ 0.95 and *aroC*, *aroA*, *aroF*, *trpG*, and *trpB*, with $r_{gCH4}$ = −0.68 to −0.74 at lower evidence $P_0$ ≥ 0.85, Fig. 4). More specifically, *trpE*, *trpD*, and *trpA*

take part in the metabolism of L-tryptophan (Fig. 4) whose catabolites (e.g. indole) are important signalling molecules in biofilm formation[78], and activation of the host immune system[79]. Moreover, 2-oxocarboxylic acid and tyrosine catabolites are precursors for the biosynthesis of coenzyme B[77,80] and methanofuran[73] methanogenic cofactors, and their diversion into the synthesis of other substrates (e.g. arginine, branched-chain amino acids or tryptophan) could explain their negative $r_{gCH4}$. Lastly, four microbial genes with negative $r_{gCH4}$ (−0.61 to −0.87, $P_0$ ≥ 0.95) were associated with methionine metabolism (*metE*, *DNMT1*) and transport (*metQ* and *metN*). Methionine is associated with minor methylotrophic methanogenesis pathway[81] in the rumen[82,83] and with an enhancement of microbial long-chain fatty acid production[84], a highly $H_2$ demanding process[44]. Our study highlights that the negative association between microbial amino acid metabolism and $CH_4$[85,86] has a host genomic

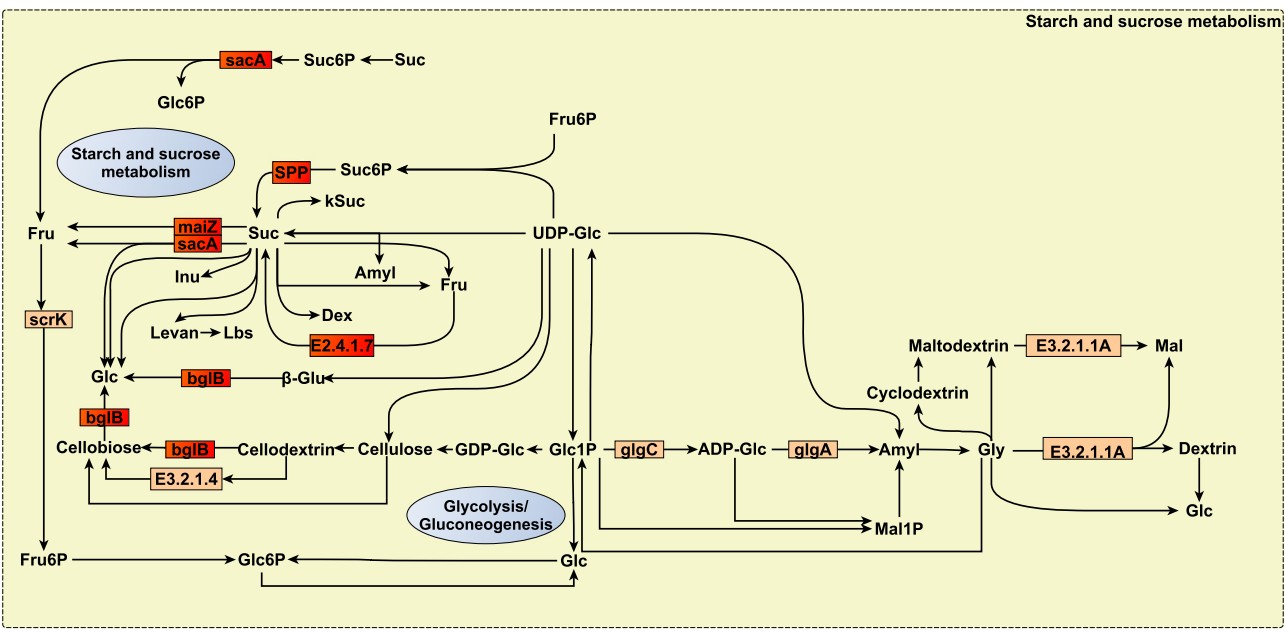

**Fig. 5 Reaction scheme of starch and sucrose metabolism in which additive log-ratio transformed microbial gene abundances strongly host genomically correlated with methane emissions ($r_{gCH4}$) are involved.** Small rectangles symbolize proteins encoded by the microbial genes. Microbial genes are highlighted in red when their $r_{gCH4}$ estimates range between −0.74 and −0.93 and show a probability of being different from 0 ($P_0$) ≥ 0.95, and in orange when they range between |0.55| and |0.77| and $P_0$ ≥ 0.85. Source data is in Supplementary Data 9. Compounds are denoted by their short names. Full names of compounds and microbial genes are given in Supplementary Data 17.

component. This could be partly due to host genomic effects[19] on ruminal feed retention times, which have opposite effects on microbial protein synthesis efficiency[67] and $CH_4$ production[87].

We obtained negative $r_{gCH4}$ (from −0.60 to −0.85, $P_0 ≥ 0.95$) for the abundance of several microbial genes responsible for sucrose metabolism (*sacA*, *maIZ*, *bgIB*, *SPP*, and *sucrose phosphorylase*, Fig. 5), including the highly abundant sucrose fermenter[88] *Eubacterium* (RA = 1.02%), transporters of multiple sugars across the membrane[85] (*ABC.MS.P1*, *ABC.MS.S*, and *ABC.MS.P*), and the microbial gene *PTS-EI* that catalyses the phosphorylation of incoming sugar substrates concomitantly with their translocation across the cell membrane. Microorganisms capable of fast growth on soluble sugars are suggested to be favoured in hosts with low rumen size and high digesta turnover rate[85,89], features also associated with low $CH_4$ emissions[87]. Degradation of easily fermentable carbohydrates, such as sucrose or starch, causes a pH decline which has a strong $CH_4$ reducing effect as a result of pH sensitivity of methanogens or $H_2$-producing microbes[90]. Furthermore, previously mentioned microbial genes *aroA* and *trpE* are involved in the shikimate pathway[91] linking sugar metabolism with the synthesis of microbial proteins (aromatic amino acids, tyrosine, phenylalanine, and tryptophan) which are an important source of amino acids for the host. Microbial protein yield from sucrose is suggested to be more persistent over time in comparison to other carbohydrates[92], and partially stored by sucrose utilizers (e.g. *Eubacterium*) for the maintenance of the microbial population[92].

We also found negative $r_{gCH4}$ for the abundance of hydrogenotrophic acetogenic bacteria *Blautia*[45], together with *Eubacterium*[93] ($r_{gCH4}$ = −0.60 and −0.73, $P_0 ≥ 0.95$), and the *fhs* microbial gene involved in the reductive Wood–Ljungdahl acetyl-CoA pathway ($r_{gCH4}$ = −0.79, $P_0 = 0.98$). Acetogens produce volatile fatty acids (mainly acetate but also propionate and butyrate[94]), which serve as host nutrients to improve animal performance[42] and simultaneously compete against methanogens for metabolic $H_2$[9,43,45]. Despite acetogenesis being thermodynamically less favourable than the reduction of $CO_2$ into $CH_4$[95] in the rumen, this may vary upon

microbial interactions and host-genomically influenced ruminal environmental factors[42,45,68]. Propionogenesis via acrylate[34,85,89,96] and lactaldehyde routes[40] was another microbial mechanism under host genomic influence lowering $CH_4$ emissions as indicated by negative $r_{gCH4}$ (−0.76 to −0.90, $P_0 ≥ 0.95$) for the abundances of microbial genes *bcd* and *pccB* involved in propanoyl-CoA metabolism and *fucO* catalysing the reduction of lactaldehyde into 1,2-propanediol, as well as the highly abundant (RA = 0.08%) lactate-producing bacteria *Kandleria* ($r_{gCH4}$ = −0.87, $P_0 = 0.99$). Propionate production from lactate not only reduces $H_2$ availability for methanogenesis[13,97] but also prevents rumen acidosis[98] and results in a more efficient rumen fermentation[99,100]. The abundance of six microbial genes encoding [4Fe-4S] cluster containing proteins (*bioB*, *cobL*, *cofG*, *nifU*, *ACO*, and *pflA*) involved in electron transfer mechanisms in redox reactions presented $r_{gCH4}$ from −0.71 to −0.87 ($P_0 ≥ 0.96$). The first two proteins are involved in the synthesis of substrates required for methanogenic cofactors; i.e. *bioB* catalyses the conversion of dethiobiotin to biotin[101], which competes with coenzyme B for the synthesis of its alkyl portion[102,103]; and *cobL* together with *hemC* ($r_{gCH4}$ = −0.91, $P_0 = 1.00$) take part in porphyrin metabolism, required for different processes including the synthesis of porphyrin-based cofactors vitamin $B_{12}$ and $F_{430}$[104]. Nitrogen fixation protein *nifU* carries out $N_2$ reduction into ammonia[105], which can act as an alternative $H_2$-consuming sink competing with ruminal methanogenesis. Further negative $r_{gCH4}$ were obtained for microbial genes in thiamine metabolism (*iscS*, *thiD*, *thiH*, and *thiE* with $r_{gCH4}$ from −0.88 to −0.70, $P_0 ≥ 0.91$)[106]; hydration of long-chain fatty acid oleate into anti-tumoral hydroxystearic acid[107,108] (*ohyA*, −0.81, $P_0 = 0.95$), or import of methanogen inhibitors long-chain fatty acids[47] (*ABCB-BAC*[109], $r_{gCH4}$ = −0.9, $P_0 = 0.99$). Moreover, highly abundant bacteria genera with ruminal fatty acid biohydrogenation activity[110,111], *Eubacterium* and *Butyrivibrio* (RA = 2.54%, $r_{gCH4}$ = −0.37, $P_0 = 0.80$) were negatively correlated with $CH_4$.

*Microbial communication and host–microbiome interaction mechanisms.* The majority of methanogens in the rumen are

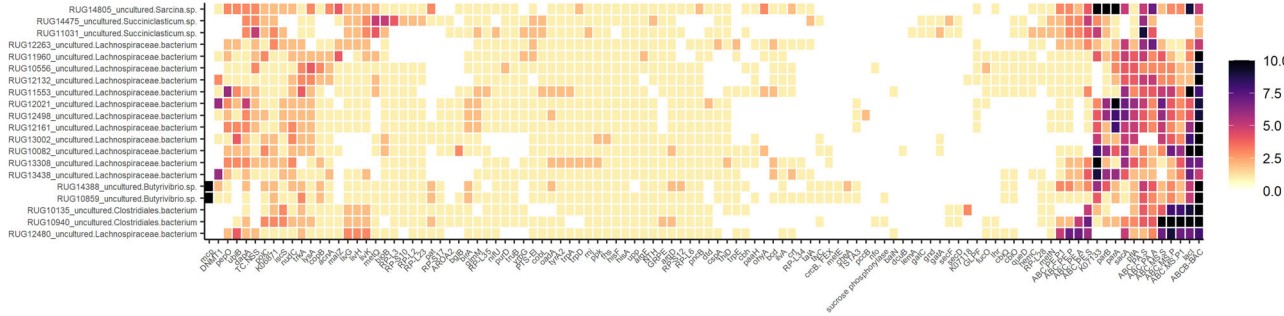

**Fig. 6 Top 20 rumen uncultured genomes (RUGs) highly enriched with the 115 microbial genes host-genomically correlated to methane emissions with a probability of being higher or lower than 0 ($P_0$) ≥ 0.95.** Colour scale represents the number of unique proteins mapping into each KEGG orthologous group (i.e. microbial gene). Source data is in Supplementary Data 11. Full names of microbial genes are given in Supplementary Data 18.

integrated into the biofilm on the surface of feed particles where $H_2$ producing bacteria are active[112–114]. We found strong negative $r_{gCH4}$ (−0.78 to −0.92, $P_0$ ≥ 0.96) for abundances of microbial genes mediating microbial interactions, involved in ABC transport of cobalt/nickel (*cbiO* and *cbiQ*) and quorum sensing-related peptide/nickel ions (*ABC.PE.P*, *ABC.PE.S*, *ABC.PE.A*, *ABC.PE.P1*) —cobalt and nickel being detrimental for hydrogenotrophic and aceticlastic methanogenic activity[48]—protein export (*secD* and *secF*) and chemotaxis (*cheA* and *mcp*); and positive $r_{gCH4}$ for transcription protein *cbpA* (0.85, $P_0$ = 0.97) acting as a microbial response to maintain plasmids replication during amino acid starvation[115]. $CH_4$ emissions were also host-genomically correlated with abundances of microbial genes mediating host-microbiome interaction; e.g. *cbh* and *baiN*[116] ($r_{gCH4}$ = −0.80, $P_0$ ≥ 0.96) involved in bacterial biosynthesis of secondary bile acids which activate metabolic receptors within gut, host liver, and peripheral tissues[116,117] and inhibit $CH_4$ production in the rumen by transferring metabolic $H_2$ into propionate production[118]. Another interesting finding is that *TSTA3*, involved in the metabolism of host-microbiome crosstalk mediator fucose[119], displays a positive $r_{gCH4}$ (0.85, $P_0$ = 0.98). Fucose is a component of mucins present in saliva[120], which is produced abundantly by ruminants and acts as a pH buffer during ruminal fermentation due to its phosphate and bicarbonate content[121]. Cellulolytic *Fibrobacter*, an indicator of high pH levels in rumen[122], was positively host-genomically correlated to *TSTA3* in our data (0.66, $P_0$ = 0.94), while lactic acid producer *Kandleria*, generally associated with low pH levels and negative $r_{gCH4}$, was host-genomically correlated to *TSTA3* negatively (−0.70, $P_0$ = 0.90). Thus *TSTA3* could be involved in signalling enhanced saliva production, resulting in increased rumen pH that is known to stimulate the growth of methanogenic archaea and $CH_4$ emissions[123].

*Genetic information processes.* Ribosomal biogenesis represented by *RP-S10*, *RP-S12*, *RP-S17*, *RP-L2*, *RP-L3*, *RP-L6*, *RP-L23*, *RP-L28*, *RP-L34*, and *RP-L35*, was one of the few microbial mechanisms with positive $r_{gCH4}$ from 0.71 to 0.84 ($P_0$ ≥ 0.95). All of them are universal ribosomal proteins homologous in bacteria, archaea, and eukarya; except for *RP-L28*, *RP-L34*, and *RP-L35* exclusively found in bacteria[124,125]. Given that protein synthesis is highly coupled with cellular growth[69], these results suggest that the rumen environment provided by low $CH_4$-emitter host genomes are related to lower growth or activities of specific microbes directly or indirectly involved in methanogenesis.

**RUGs enriched with $CH_4$-related microbial genes are strongly host-genomically correlated to $CH_4$ emissions.** The 20 highly prevalent (present in >200 animals) RUGs containing the highest

number of unique proteins from the 115 microbial genes with strong $r_{gCH4}$ were all bacterial RUGs carrying between 114 to 180 unique proteins classified into 60 to 84 microbial genes (Fig. 6 and Supplementary Data 11 and 12). Of these 20 highly enriched bacterial RUGs, 18 showed negative $r_{gCH4}$ consistently with the majority of the microbial genes; 6 of them with $r_{gCH4}$ < −0.65 ($P_0$ ≥ 0.85) from which 5 RUGs were classified as uncultured *Lachnospiraceae bacterium* (RUG10082, RUG13438, RUG13308, RUG13002, RUG12132) and 1 as uncultured *Clostridiales bacterium* (RUG10940). The abundance of *Blautia* and *Dorea* microbial genera within *Lachnsopiraceae* family (identified by alignment to Hungate1000 collection and Refseq databases) also presented negative $r_{gCH4}$ < −0.72 ($P_0$ ≥ 0.95, Supplementary Data 7). We also investigated the enrichment of these 115 microbial genes in the 6 RUGs with $r_{gCH4}$ ($P_0$ ≥ 0.95) annotated at the genus level (Supplementary Data 8), and in those RUGs annotated in the same phylogeny level as any of the 29 microbial genera with $r_{gCH4}$ ($P_0$ ≥ 0.95, Supplementary Data 7), which had low occupancies in our cattle population (<200 animals) and therefore were not included in the 225 considered for breeding (see methods). Our findings show that part of the mechanisms identified in this study occurs in the 5 RUGs classified as uncultured *Methanobrevibacter* strains, each carrying at least 45 out of the 115 microbial genes (Supplementary Fig. 3). The uncultured *Methanobrevibacter* strain with positive $r_{gCH4}$ (RUG12982) carried fewer unique proteins (67 vs. 75 to 93) and microbial genes (51 vs. 55 to 62) than the other 4 uncultured *Methanobrevibacter* sp. RUGs with negative $r_{gCH4}$; lacking, for example, *argD* in arginine biosynthesis, *tyrA2* in tyrosine and tryptophan metabolism, and *DNMT1* in methionine metabolism, which reinforces the hypothesis of functional versatility amongst different *Methanobrevibacter* strains explaining their different effects and estimated $r_{gCH4}$ on $CH_4$ emissions. Low-occupancy RUGs annotated as *Eubacterium ruminatum*, *Eubacterium pyruvativorans*, *Kandleria vitulina*, and uncultured sp. of *Blautia* RUGs carried at least 49 out of the 115 microbial genes each (Supplementary Fig. 3). Interestingly, their counterparts identified at the genera level presented negative $r_{gCH4}$ < −0.60 ($P_0$ ≥ 0.95, Supplementary Data 7).

**Microbiome-driven breeding of the bovine host for mitigation of $CH_4$ emissions.** The comprehensive findings of the host-genomic associations between microbial genus/RUG/gene abundances and $CH_4$ emissions enabled us to predict its mitigation potential when applying genomic selection targeting each of them individually (Supplementary Data 13), indirectly informing about the impact of each microbial mechanism on methanogenesis. Considering 30% of our cattle population being selected based on the abundances of each microbial gene, *maiZ* in sucrose

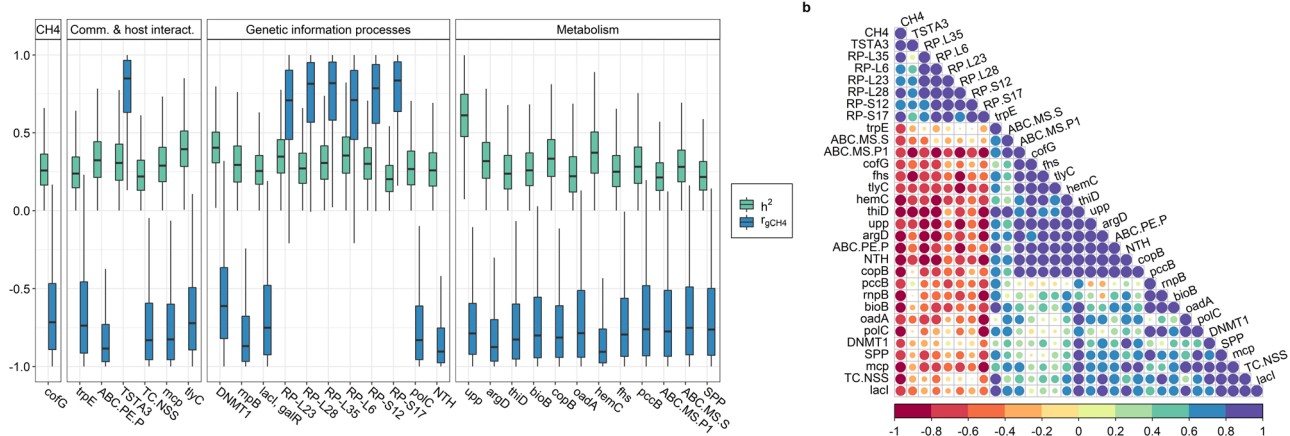

**Fig. 7 Microbial genes selected to be used collectively for selecting the host genomes associated with low CH₄ emissions, meeting 3 criteria: showing significant heritability ($h^2$) based on Bayes Factor >3 and Deviance Information Criterion difference between models with or without host genomic effects $\leq -20$; a host genomic correlation with CH₄ ($r_{gCH4}$) with a probability of being higher or lower than 0 ($P_0$) > 0.95, and showing a relative abundance >0.01%. a** Estimates of $h^2$ and $r_{gCH4}$ (error bars represent the highest posterior density interval enclosing 95% probability). Microbial genes grouped by microbial biological processes: methane metabolism (CH₄), microbial communication and host–microbiome interaction (Comm. & host interact.), genetic information processes and metabolism other than CH₄ (Metabolism). **b** Correlogram showing the median of the pairwise host-genomic correlations estimates among the additive log-ratio transformed microbial gene abundances selected for breeding purposes. Source data is in Supplementary Data 14. Full names of microbial genes selected for breeding purposes are given in Supplementary Data 19.

metabolism, *ABC.PE.P* in quorum sensing peptide/nickel transport, *hemc* in porphyrin or *upp* in pyrimidine metabolism are predicted to result in the highest CH₄ mitigation potential (−5.2, −5.3, −5.8 and −6.54% of CH₄ emissions mean respectively, $P_0 \geq 0.99$). Subsequently, our study aimed to find a group of heritable (BF > 3 and Deviance Information Criterion difference between models with or without host genomic effects $\leq -20$) ruminal microbial genera/RUGs/genes (RA > 0.01%) with strong $r_{gCH4}$ ($P_0 \geq 0.95$) to be used collectively for selecting the host genomes associated with low CH₄ emissions. We identified 2 microbial genera (*Eubacterium* and *Blautia*), 3 RUGs (two annotated as uncultured *Methanobrevibacter* sp. and one as uncultured *Prevotellaceae bacterium*) and 38 microbial genes meeting these requirements (Supplementary Data 14). We selected 30 out of the 38 microbial genes (Fig. 7a) covering several microbial mechanisms, e.g. sugar and nickel transport (*ABC.PE.P*, *ABC.MS.P1* and *ABC.MS.S*), fucose sensing (*TSTA3*), chemotaxis (*mcp*), ribosomal biosynthesis (*RP-L6*, *RP-L23*, *RP-L28*, *RP-L35*, *RP-S12* and *RP-S17*), reductive acetogenesis (*fhs*) and metabolism of amino acids (*argD*), sucrose (*SPP*), CH₄ (*cofG*), biotin (*bioB*), propionate (*pccB*), porphyrin (*hemC*), thiamine (*thiD*) and pyrimidine (*upp*). A deep study of the host-genomic correlations among these 30 selected microbial genes showed a common host genomic background influencing the abundance of *ABC.PE.P*, *ABC.MS.P1*, *fhs*, *cofG*, *argD*, *hemC*, *thiD*, *upp*, *tlyC*, *NTH*, and *copB* with host-genomic correlations among each other ranging from 0.62 ($P_0 = 0.90$) to 0.99 ($P_0 = 1.00$) (Fig. 7b).

Finally, we evaluated the accuracies and response to selection in CH₄ emission mitigation in our population based on the prediction of CH₄ host genomic effects using three different sources of information: (1) CH₄ emissions measured by the "gold-standard" technique of respiration chambers, (2) the 30 microbial gene abundances exhibiting strong $r_{gCH4}$, and (3) combining both preceding criteria. A single (1) or multiple (2, 3) trait genomic estimation approach was applied in each case. In (2) and (3), CH₄ host genomic effects were estimated based on observations of the 30 microbial gene abundances, the genomic relationship matrix amongst individuals, and the estimated host-genomic and residual (co)variance matrix comprising CH₄ and the 30 microbial gene abundances; assuming unknown (2) or

known (3) CH₄ observations (see methods). Using microbiome-driven breeding based on the abundance of 30 specific microbial genes (2) resulted in the mean estimation accuracy of host genomic effects for CH₄ emissions to be 34% higher than using measured CH₄ emissions (1) (0.70 ± 0.18 vs. 0.52 ± 0.11) and confirmed that functional microbial genes are an extremely valuable source of information to perform host genomic evaluations for CH₄ emissions. Using the combined selection criteria (3), the accuracy of estimation was 14% larger than using rumen microbial gene information alone (0.80 ± 0.20). Response to selection in CH₄ emissions achieved by selecting animals with low CH₄ emission host genomic effects predicted exclusively by microbial gene abundance information resulted in a reduction in emissions of −1.43 ± 0.14 to −3.32 ± 0.77 g CH₄/kg DMI per generation, depending on selection intensity (from 1.16 to 2.67 in the analysed population, Fig. 8 and Supplementary Data 16). These results indicate that in our population, microbiome-driven breeding for CH₄ emissions reduced its magnitude by 7–17% of its mean per generation, without the necessity for costly measures of CH₄ emissions.

**Robustness of estimation of genomic parameters of CH₄ emissions and microbiome traits from a cross-classified design of breeds and basal diets.** The data, from the highly environmentally standardized experiments, comprised of animals from different breeds that were offered different diets, which could be challenging as different breeds might have different genomic backgrounds for the analysed traits, and different diets could have inflated their variances. To consider the difference in means of these effects we fitted a combination of experiment, breed, and diet effects so that an adjustment of each of these effects and their interactions was achieved. To analyse whether after this adjustment the genomic variances of CH₄ emissions and the abundances of 30 microbial genes selected for microbiome-driven breeding are homogeneous across breeds and diets, we computed their posterior distributions separately (Supplementary Figs 4 and 5), following the partition of the variance suggested by Sorensen et al.[126] and recently used by Lara et al.[127]. Using this methodology, we found that our genomic parameter estimates for CH₄ emissions are based on similar genomic variances across breeds

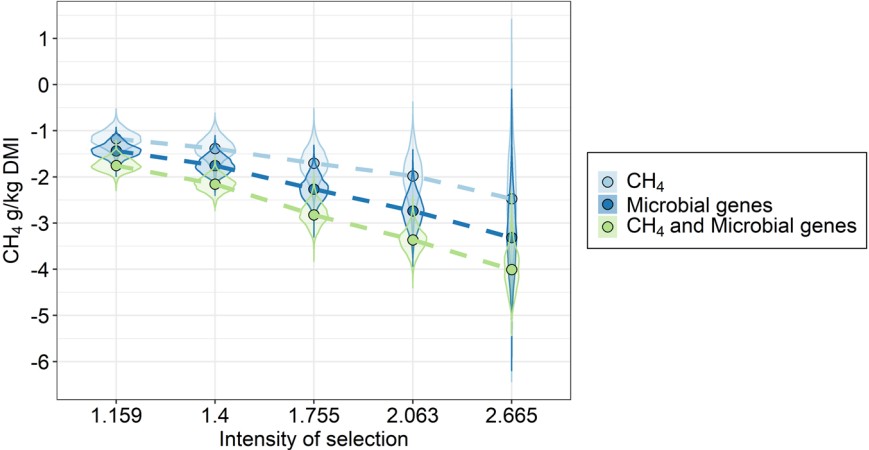

**Fig. 8 Response to selection per generation on methane (CH₄) emissions estimated using direct genomic selection based on measured CH₄ emissions (light blue), indirect genomic selection based on 30 microbial gene abundances most informative for host genomic selection for CH₄ (dark blue) or selection on both criteria (green).** Intensities of selection 1.1590, 1.400, 1.755, 2.063, or 2.665 are equivalent to selecting 30, 20, 10, 5, or 1%, respectively, of our $n = 285$ animal population with CH₄ and metagenomic data based on the above-described selection criteria. Dots display the medians and violin plots represent the estimated marginal posterior distributions of the response to selection for each intensity of selection and breeding strategy. Source data is in Supplementary Data 16.

(with medians of 3.8, 3.7, 4.0, and 3.9 (g/kg DMI)$^2$ for Aberdeen Angus, Limousin, Charolais crosses and pure breed Luing, respectively) and diets (with medians of 3.8 and 3.9 (g/kg DMI)$^2$ for Forage and Concentrate based diets) with almost entirely overlapping distributions indicating their homogeneity (Supplementary Fig. 4). We also identified homogeneous genomic variances for the abundances of microbial genes with overlapping distributions across breeds (Supplementary Fig. 5a) and diets (Supplementary Fig. 5b); for example, we estimated genomic variances with medians between 0.024–0.028 (g/kg DMI)$^2$ across breeds and 0.028–0.029 (g/kg DMI)$^2$ across diets for *RP-L35*, or between 0.25–0.30 (g/kg DMI)$^2$ across breeds and 0.292–0.296 (g/kg DMI)$^2$ across diets for *cofG* (Supplementary Fig. 5). These results indicate that the data recorded under controlled experimental conditions with a cross-classified breed and diet two-way experimental design and progeny groups balanced over diets resulted in reliable and robust genomic parameters estimates.

## Discussion
Previous metagenomic studies using 16S rRNA identified microbiota revealed host genomic effects on the rumen microbial community[7,18–25], e.g., Wallace et al.[21] found significant heritabilities for several members of *Prevotella* and *Butyrivibrio* genera. These species, together with other non-heritable OTUs from the core microbiota, explained up to 40% of the phenotypic variation of CH₄ emissions[21] that is expected to be the result of both genetic and environmental correlations, which we estimated in the present study separately for each taxa and microbial gene. Moreover, the challenge of previous studies is the lack of taxonomic resolution to sufficiently explain these associations[21], which we resolved using metagenome-assembled RUGs. For example, we revealed that different *Methanobrevibacter* strains expressed divergent host-genomic correlations with CH₄ emissions which correspond to the diverse microbial gene content of these strains.

To increase the predictability of CH₄ emissions, Roehe et al.[7], suggested genome-resolved metagenomics and identified 20 microbial gene abundances, mainly involved in the methanogenesis pathway, which explained 81% of the phenotypic variation in the emissions. In contrast, the present study uses a host genomic-microbiome analysis strategy and provides a robust and

comprehensive insight into joint host-genomic correlations between rumen microbial genes known to affect complex functional mechanisms and CH₄ emissions which further enabled us to predict the expected response of selection in the emissions based on each and a combination of the microbial genes. The findings of this research will be of major importance for the mitigation of the highly potent GHG CH₄ in bovine through genomic selection on the functional microbiome associated with CH₄ referred to as microbiome-driven breeding. The highlights of our research are that the host genome influences CH₄ emissions by favouring the growth of reductive acetogenic microbes limiting the excess of metabolic H₂ substrate (specifically, *Blautia*, *Eubacterium* genera and microbial gene *fhs* found in the genome of uncultured *Lachnospiraceae bacterium*, *Eubacterium pyruvativorans*, *Eubacterium ruminatum*, uncultured *Eubacterium* and *Blautia sp.* RUGs); and promoting the shift in the fermentation towards volatile fatty acids (*Kandleria* genera and microbial genes *bcd*, *pccB*, *fucO*, carried by uncultured *Lachnospiraceae bacterium* and *Kandleria vitulina* RUGs) and microbial proteins yield including arginine and branched-chain amino acids (*argF*, a*rgD*, *ilvA*, *AROA2*), tryptophan, tyrosine and phenylalanine (*trpA*, *trpD*, *trpE*, *tyrA2*, *paaH*) and methionine (*metE* or *DNMT1*), which are expected to lead to animals with improved efficiency of converting feed into nutrients[42,128]. Moreover, host genome contributes to lower CH₄ emissions by enhancing the growth of microbes that consume H₂ in alternative pathways (e.g. nitrogen fixation (*nifU*)); by promoting the pathways that divert specific substrates (e.g. tyrosine and 2-oxocarboxylic acid catabolites) required to produce methanogenic coenzymes or cofactors (coenzyme B and methanofuran) to other routes; and that inhibit methanogenic organisms (e.g. by the presence of branched-chain amino acids or cobalt/nickel (*cbiO*, *cbiQ*)) and maintain a lower optimum ruminal pH (sucrose metabolism (*sacA*, *maIZ*, *bgLB*, *SPP*, *sucrose phosphorylase*)) preventing gut disorders (e.g. thiamine metabolism (*iscS*, *thiD*, *thiH*, *thiE*)). The latter result supports our hypothesis that hosts who are genomically resilient to gut disorders produce less CH₄, which is compatible with nutritional studies demonstrating that blocking methanogenesis has no undesirable effects on cattle health status or feed intake[86]. A further highlight of our study is that the host genome influenced the ABC transport of different metabolites (some of them in quorum sensing processes (*ABC.PE.P*,

*ABC.PE.S*, *ABC.PE.A*, *ABC.PE.P1*)), interspecies electron transfers (*bioB*, *cobL*, *cofG*), sensitivity of environmental conditions (*cheA*, *mcp*), and host–microbiome interaction mechanisms (*cbh*, *baiN*, *TSTA3*), all host-genomically associated with CH$_4$ emissions. These results shed light on the complex processes of methanogenesis regulated by different microbial mechanisms where communication between microbial communities and their interactions with the host plays an essential role. Genetic information processes in the microbiota (e.g. ribosomal biosynthesis (e.g. *RP-S10*)) also had a substantial host genomic effect on CH$_4$ emissions, potentially reflecting different microbial community growth profiles.

Our findings on the functional microbial level are complementary to studies investigating the biological mechanisms underlying host genome influence on the colonization and maintenance of specific ruminal microbial groups, such as host genomic effects on rumen size[87], muscle contraction associated with passage rate[19], or ruminal pH[20]. Other studies in bovines have elucidated host candidate genes for CH$_4$ emissions involved in similar mechanisms[87,129,130], fitting into our demonstrated hypothesis that the host genome commonly influences rumen microbiome profile and CH$_4$ emissions. From nutritional studies[66], it is well known that the rumen pH has an overarching effect on the rumen microbial community and its metabolism. The rumen pH is intimately related to the production level of buffer-acting saliva[98] that is rich in fucose[120]. We found that the abundance of *TSTA3*, encoding the sensor for host–microbiome crosstalk mediator fucose[119], was host-genomically positively correlated to CH$_4$ emissions and to *Fibrobacter* genera, an indicator for high pH[122], making *TSTA3* a highly valuable biomarker for rumen pH, CH$_4$ metabolism and potentially, for host–microbiome mediation to enhance saliva production.

Our results provide comprehensive insight into which communities and functions of the rumen microbiome can be modified by genomic selection to obtain low CH$_4$-emitter animals. We revealed that specific microbiome functionalities (i.e. microbial gene abundances) are more informative for breeding purposes than specific taxonomies, as indicated by a higher number of microbial genes than genera/RUGs being host-genomically correlated to CH$_4$ emissions. This could be due to the closely defined function of those genes, e.g. being involved in producing specific substrates or mediating a specific pathway that interferes with CH$_4$ metabolism; while each microbial genera expresses many microbial gene functions as indicated by functional versatility within different niche-specific species or clades classified in the same genus[12,34,89,131,132] (as observed within different RUGs annotated as uncultured *Methanobrevibacter* strains); or as a result of horizontal transfer of genes among microbial species[133,134]. Thus, the knowledge generated in this study overcomes previous efforts exploring the breeding possibilities of rumen microbiome, where overall functional description was not considered and the number of taxonomic units associated with CH$_4$ was limited[21]. Our previous research has shown that the abundance of microbial communities, in particular their genes and interactions, are excellent biomarkers for the phenotypic prediction of CH$_4$ emissions[7,10,37]; however, the data sets were insufficient to estimate host genomic influence on these biomarkers[7]. The present study using a unique data set with highly standardized hosting and management conditions represents a large step further by discovering 38 heritable microbial gene abundances strongly host-genomically correlated with CH$_4$ emissions and by designing a microbiome-based breeding strategy to evaluate their potential to mitigate CH$_4$. Microbiome-driven (indirect) genomic selection for CH$_4$ emissions collectively using 30 of these microbial gene abundances resulted in our population in substantial mitigation of CH$_4$ (up to 17% of its

mean per generation; approximately 8% per year using genomic selection), even larger than direct genomic selection based on the accurately measured CH$_4$ emissions. This mitigation potential is permanent and can be cumulatively increased over generations. The selection strategy would at least partially avoid the high cost involved in measuring CH$_4$ emissions, and the cost-effectiveness of indirect selection could be further improved by the development of a microarray to quantify the abundances of the most informative microbial genes[135]. Another advantage of the proposed selection strategy is that it is based on host-genomic correlations between microbial gene abundances and CH$_4$ emissions which as we discussed have specific biological meanings.

## Methods

**Animals**. Animal experiments were conducted at the Beef and Sheep Research Centre of Scotland's Rural College (SRUC). The experiment was approved by the Animal Experiment Committee of SRUC and was conducted following the requirements of the UK Animals (Scientific Procedures) Act 1986. The data were obtained from 363 steers used in different experiments[38,39,136–138] conducted over 5 years (2011, 2012, 2013, 2014, and 2017) in the same farm under the same hosting conditions. In these experiments, we tested different breeds (rotational cross from Aberdeen Angus and Limousin breeds, Charolais-crosses, and pure breed Luing) and two basal diets consisting of 480:520 and 80:920 forage: concentrate ratios (DM basis) and subsequently referred to as forage and concentrate diet. Supplementary Data 15 gives the distribution of the animals across experiments, breeds, and diets. Each experiment was balanced for the breed and diet effects, as well as the progenies of each sire was balanced over each diet, so that the experimental design by itself has high power to disentangle genomic effects from diet effects. Additionally, a power analysis indicated that for the given number of animals per experiment, a genetic design of sires with on average 8 progeny per sire showed the highest power to identify genetic differences between sires in methane yield with an achieved power of 0.93 using the sire estimates and root mean square error of 3.27 as obtained in experiment of our previous study[7].

**Methane emissions data**. Methane emissions were individually measured in 285 of the 363 animals for 48 h within six indirect open-circuit respiration chambers[39]. One week before entering the respiration chambers, the animals were housed individually in training pens, identical in size and shape to the pens inside the chambers, to allow them to adapt to being housed individually. At the time of entering the chamber, the average age of the animals was 528 ± 38 days and the average live weight was 659 ± 54 kg. In each experiment, the animals were allocated to the respiration chambers in a randomized design within breed and diet. Animals were fed once daily, and the weight of the feed offered and refused was recorded. Methane emissions were expressed as g of CH$_4$/kg of DMI, by dividing the average CH$_4$ emissions (g/day) by the average DMI (kg/day) recorded both over 48 h.

**Hosts genomic samples**. For host DNA analysis, 6–10 ml of blood from the 363 steers were collected from the jugular or coccygeal vein in live animals or during slaughter in a commercial abattoir. In addition to the 363 samples, 7 blood and 23 semen samples from sires of the steers were available (n = 393 samples in total). Blood was stored in tubes containing 1.8 mg EDTA/ml blood and immediately frozen to −20 °C. Genomic DNA was isolated from blood samples using the Qiagen QIAamp toolkit and from semen samples using Qiagen QIAamp DNA Mini Kit, according to the manufacturer's instructions. The DNA concentration and integrity were estimated with Nanodrop ND-1000 (NanoDrop Technologies). Genotyping was performed by Neogen Genomics (Ayr, Scotland, UK) using GeneSeek Genomic Profiler (GGP) BovineSNP50k Chip (GeneSeek, Lincoln, NE). Genotypes were filtered for quality control purposes using PLINK version 1.09b[139]. Single Nucleotide Polymorphisms were removed from further analysis if they met any of these criteria: no known chromosomal location according to Illumina's maps[140], non-autosomal locations, call rates less than 95% for SNPs, deviation from Hardy–Weinberg proportions ($\chi^2$ test $P < 10^{-4}$), or minor allele frequency (MAF) <0.05. Seven animals, showing genotypes with a call rate <90%, were removed so that 386 animals and 36,780 autosomal SNPs remained for the analyses.

**Hosts metagenomic samples**. For microbial DNA analysis, post mortem digesta samples (approximately 50 ml) from 363 steers were taken at slaughter immediately after the rumen was opened to be emptied. Five ml of strained ruminal fluid was mixed with 10 ml of PBS containing glycerol (87%) and stored at −20 °C. DNA extraction from rumen samples was carried out following the protocol from Yu and Morrison[141] based on repeated bead beating with column filtration and DNA concentrations and integrity was evaluated by the same procedure (Nanodrop ND-1000) as for blood samples. Four animals out of 363 did not yield rumen samples of sufficient quality for metagenomics analysis. DNA Illumina TruSeq libraries were

prepared from genomic DNA and sequenced on Illumina HiSeq systems 2500 (samples from 4 animals from the experiment year 2011), HiSeq systems 4000 (samples from 284 animals from experiment years 2011, 2012, 2013 and 2014)[8,34] or NovaSeq (samples from 76 animals from the experiment year 2017) by Edinburgh Genomics (Edinburgh, Scotland, UK). Paired-end reads (2 × 100 bp for Hiseq systems 2500 and 2 × 150 bp for Hiseq systems 400 and NovaSeq) were generated, resulting in between 7.8 and 47.8 GB per sample (between 26 and 159 million paired reads).

**Bioinformatics.** For phylogenetic annotation of rumen samples, the sequence reads of 359 samples were aligned to a database including cultured genomes from the Hungate 1000 collection[40] and Refseq genomes[41] using Kraken software[142]. From 1178 cultured microbial genera identified, we used only those present in all the samples and with a RA > 0.001% (1108 microbial genera) for downstream analysis, equivalent to 99.99% of the total number of counts. We used the 4941 RUGs generated by Stewart et al.[34] with sequences of 282 rumen samples included in this study to identify and quantify the abundance of uncultured species. A detailed description of the metagenomics assembly and binning process and estimation of the depth of each RUG in each sample is described in Stewart et al.[34]. For breeding purposes, microbial taxa that are present in a large proportion of the animals are required; so we discarded those RUGs present in <200 animals (using a cut-off of 1× coverage) and kept 225 RUGs. RUGs coverages <1, which comprised 17.7% of the whole RUGs data set were imputed based on a Geometrical Bayesian-multiplicative method (GBM) of replacement by using *cmultrepl* function in zCompositions package[143]. This algorithm imputes zero values from a posterior estimate of the multinomial probability assuming a Dirichlet prior distribution with default parameters for GBM method[144] and performs a multiplicative read-justment of non-zero components to respect original proportions in the composition. The 225 RUGs considered showed a mean relative abundance ≥0.15%. Bioinformatic analysis for the identification of rumen microbial genes was carried out as previously described by Wallace et al.[145]. Briefly, to measure the abundance of known functional microbial genes whole metagenome sequencing reads were aligned to the Kyoto Encyclopaedia of Genes and Genomes (KEGG) database (https://www.genome.jp/kegg/ko.html)[146] using Novoalign (www.novocraft.com). Parameters were adjusted such that all hits were reported that were equal in quality to the best hit for each read and allowed up to a 10% mismatch across the fragment. The KEGG orthologous groups (KO) of all hits that were equal to the best hit were examined. If we were unable to resolve the read to a single KO, the read was ignored; otherwise, the read was assigned to the unique KO, the resulting KO grouping corresponding to a highly similar group of sequences. We identified 3,602 KO (also referred to as microbial genes), common in all animals. As for microbial genera, we used only core microbial genes present in all the samples and with a RA > 0.001% (1142 microbial genes) for downstream analysis, equivalent to 96.25% of the total number of counts. We combined information from KEGG, UniProt, and Clusters of Orthologous Groups of protein databases to classify 1141 microbial genes into classes depending on the biological processes they are involved in $CH_4$ metabolism (34), metabolism other than $CH_4$ pathway (511), genetic information processes (329), microbial communication and host-microbiome interaction (207) and other unknown or at present poorly characterized (61).

**Statistics and reproducibility**

*Log-ratio transformation of metagenomic data.* To describe the composition of the microbiome at the taxonomic level (cultured microbial genera and RUGs) and functional level (KO or microbial genes) we estimated their RA by dividing each microbial genus/gene (in counts) by the total sum of counts of microbial genera/genes identified in each sample (Supplementary Data 1–3). To compute host genomic analysis on the microbial cultured genera and gene abundances, we first applied a log-ratio transformation to attenuate the spurious correlations due to their compositional nature[147]. We used additive log-ratio transformation by using a reference microbial genera/gene because of the linear independence achieved between each variable and all the variables in the composition and because of the facility of its interpretation[148,149]. Assuming J denotes the number of variables in each microbial database (J = 1142 for microbial genes and 1108 for cultured microbial genera), and J−1 all of them excluding the reference microbial genera/gene, the RA of each microbial genus/gene within a sample was transformed as follows[150]:

$$\ln\left(\frac{x_j}{x_{ref}}\right) = \ln(x_j) - \ln(x_{ref}), j = 1, \dots, J-1, j \neq ref \quad (1)$$

where $x_j$ is the RA of each microbial genus/gene j and $x_{ref}$ is the RA of a specific microbial genus/gene in the database selected as a reference. We selected the 16S rRNA gene and *Oribacterium* as reference microbial gene and microbial genus, respectively. These reference variables were selected based on the criteria recommended by Greenacre et al.[151]: (1) present in rumen samples of 359 animals; (2) highly abundant (mean RA 8.56% and 0.35%, respectively); (3) not mentioned to be associated with $CH_4$ emissions in previous literature; (4) low log-ratio variance so the variation mainly proceeds to the numerator (0.09 and 0.24, both located in the first quartile when ordering the microbial variables by log-ratio variance in decreasing order) and (5) reproducing the geometry of the full set of log-ratios in the original data set shown by the estimate of the Procrustes correlation[148,152]

between the geometrical space defined by all log-ratios and the one defined by the selected additive log-ratios (Procrustes correlation is 0.95 and 0.92). *Oribacterium* is a strictly anaerobic and non-spore-forming bacterial genus from the order Clostridiales and family of *Lachnospiraceae*; commonly found in the rumen of cattle[19,153] and also in the human oral cavity[154,155]. The abundance of RUGs was centred log-ratio transformed[149] as additive log-ratio transformation was here hampered by the difficulty of selecting a reference RUG present in all animals. Assuming J denotes the total number of RUGs (J = 225):

$$\ln\left(\frac{x_j}{[\prod_j x_j]^{\frac{1}{J}}}\right) = \ln(x_j) - \frac{1}{J}\sum_j \ln(x_j), j = 1, \dots, J \quad (2)$$

where $x_j$ is the depth of each RUG j.

*Influence of breed, diet, and experiment on $CH_4$ emissions and microbial traits.* In our genomic models, we applied an optimal adjustment of the fixed effects as a combination of experiment, breed, and diet so that adjustment of each of these effects and their interactions was achieved (see below section: Estimation of host genomic parameters of $CH_4$ emissions and microbial traits). Sequencer effects were except for only 4 samples nested within the experiment and therefore were accounted for due to the inclusion of this effect. Since these are nuisance effects with the potential of influencing the estimation of genomic effects, we carried out further exploratory analyses and revealed that they were appropriately adjusted and do not interfere with the estimation of genomic parameters. In this exploratory analyses, we evaluated the effect of breed, diet, and experiment on microbial genes, genera, and RUGs using a PERMANOVA analysis with 999 permutations computed with the R package vegan[156]. Diet and experiment showed the largest effects on the composition of the rumen microbiome. Diet accounted for 18.9, 12.0, and 13.19% of the total phenotypic variance in the microbial genes/microbial genera/RUGs databases, while experiment explained 7.5, 11.17, and 6.09%, respectively (P-value < 0.001 in all cases). Breed effects on microbial genes/microbial genera/RUGs databases were depreciable (0.49, 0.65, and 0.78%) and non-significant (P-values = 0.668, 0.385, and 0.141) indicating that the breeds considered in this study did not show substantial differences in their functional or taxonomical microbiome composition. As expected, all effects became negligible (explaining 0% of the phenotypic variance) after their adjustment as a combined fixed effect. We also evaluated the effects of breed and diet on $CH_4$ emissions mean and variance (Supplementary Tables 1 and 2). Before adjustment, diet had a significant (P-value = $2.2 \times 10^{-16}$) large effect on $CH_4$ (explained 41.2% of the phenotypic variance), while the effect of breed (2.92%, P-value = 0.0007) was smaller but still significant, however, both turned to 0% after the fixed-effects adjustment. Importantly, for the reliability of estimation of genetic parameters, animals from different breeds, or fed different diets presented homogenous $CH_4$ phenotypic variances (P-values of Levene test = 0.19 and 0.48).

*Estimation of host genomic parameters of $CH_4$ emissions and microbial traits.* Genomic heritabilities ($h^2$) of $CH_4$ emissions, log-transformed microbial genera (n = 1107), RUGs (n = 225) and microbial genes (n = 1141) abundances were estimated by fitting 2474 GBLUP univariate animal models described as:

$$y = Xb + Zg + e \quad (3)$$

Data were assumed to be conditionally distributed as:

$$y\,|\,b,g,R \sim N(Xb + Zg, I\sigma_e^2), \quad (4)$$

where y is the n = 359 or n = 285 observations of the microbiome or $CH_4$ emissions trait, b is the vector of fixed effects including a combination of breed, diet, and experiment effect, g is the random host genomic effect, e is the residual of the model, and X and Z are known incidence matrices for fixed and random effects. Host genomic effects were normally distributed as:

$$g\,|\,G_{RM}, \sigma_g^2 \sim N(0, G_{RM}\sigma_g^2) \quad (5)$$

Residuals were independently normally distributed as:

$$e\,|\,I, \sigma_e^2 \sim N(0, I\sigma_e^2), \quad (6)$$

in which $\sigma_g^2$ and $\sigma_e^2$ are the host-genomic and residual variances, I is an identity matrix of the same order as the number of data, and $G_{RM}$ is the host-genomic relationship matrix between the individuals defined as[157]:

$$G_{RM} = \frac{W'W}{2\sum_n^1 p_n(1-p_n)}, \quad (7)$$

where W contains genotypes adjusted for allele frequency, and $p_n$ is the allele frequency for marker n in the whole genotyped population. Host genomic and residual effects were assumed to be uncorrelated between them. Host genomic ($r_{gCH4}$) and residual correlations among $CH_4$ emissions and log-transformed abundances of microbial genera, RUGs and microbial genes were estimated by fitting 2473 GBLUP bivariate animal models including the same effects as Eq. (3). Host genomic effects were distributed as:

$$g\,|\,G_{RM}, G_0 \sim N(0, G_{RM} \otimes G_0), \quad (8)$$

and residuals as:

$$R = e|R_0 \sim N(0, I \otimes R_0), \qquad (9)$$

where $G_0$ and $R_0$ are the $2 \times 2$ host genomic and residual (co)variance matrices between $CH_4$ emissions and each microbial genus, RUG, or microbial gene, $I$ is an identity matrix of the same order as the number of individuals with data. Bayesian statistics were used[158], assuming priors for all unknowns as implemented in THRGIBBSF90 program[159]. Results were based on Markov chain Monte Carlo chains consisting of 1,000,000 iterations, with a burn-in period of 200,000, and to reduce autocorrelations only 1 of every 100 samples was saved for inferences. In all analyses, convergence was tested using the POSTGIBBSF90[159] program by calculating the Z criterion of Geweke (varying between −0.05 and 0.05 in univariate and −0.09 and 0.1 in bivariate models). Monte Carlo sampling errors were computed using time-series procedures and checked to be at least 10 times lower than the standard deviation of the marginal posterior distribution. As $h^2$ estimates, we used the median of its marginal posterior distribution of $CH_4$, each microbial genus, RUG, or microbial gene, and the highest posterior density interval at 95% probability ($HPD_{95\%}$). We considered microbial abundances with $h^2$ estimates <0.20 being lowly heritable, $0.20 < h^2 < 0.40$ being moderately heritable and $h^2$ estimates >0.40 being highly heritable. To test the significance of host genomic effects we analysed the fitness of the full univariate genomic model vs. the univariate model without host genomic effect by comparing the deviance information criterion (DIC)[160] between models and computed the BF using an approximation of the marginal likelihood probability[161]. BF was corrected for multi-testing by assuming prior odds ratios equal to 1/number of hypothesis tests performed ($n = 2473$). We assumed evidence of a host genomic effect on the microbial trait when the DIC of the full model was at least 20 points lower than the DIC in the reduced model, and corrected BF was >3[162]. To study the homogeneity of genomic variances for $CH_4$ emissions and microbial traits within each breed and within each diet, we computed the marginal posterior distribution of the genomic variance for the most relevant microbial traits in each of the 4 different breeds and in each of the 2 different diets, following the same steps as described for the partition of the variance in Sorensen et al.[126,127]. As an estimate for the host genomic correlations, we used the median of its marginal posterior distribution and the $HPD_{95\%}$. To investigate the confidence level of $r_{gCH4}$, we estimated the posterior probability of $r_{gCH4}$ being >0 when the median of the correlation was positive or <0 when the median was negative ($P_0$). We only considered significant those $r_{gCH4}$ estimates with ($P_0$) ≥ 0.95. Additionally, univariate analyses were run using the frequentist approach using AIREMLF90[159] and similar results were obtained.

To predict the impact of indirect selection for reduced $CH_4$ emissions using microbial genera/genes significantly ($P_0 \geq 0.95$) host-genomically correlated with $CH_4$ emissions, we estimated the marginal posterior distribution of the correlated response in $CH_4$ emissions after host genomic selection for each of these microbial genera/genes, considering only the own performance of each individual[163]:

$$R_{CH4j} = i \, h_j \, r_{gCH4j} \sigma_{gCH4}, \qquad (10)$$

where $R_{CH4j}$ presents the selection response in $CH_4$ emissions after selection for the abundance of each microbial genus/gene $j$, $i$ is the intensity of selection considered to be 1.159 (equivalent to 30% of our cattle population being selected based on the selection criterion), $h_j$ is the marginal posterior distribution of the square root of the $h^2$ estimate of the microbial genus/gene from univariate analyses, $r_{gCH4j}$ is the marginal posterior distribution of the host-genomic correlation between $CH_4$ emissions and microbial genus/gene $j$ from bivariate models, and $\sigma_{gCH4}$ is the squared root of the genomic variance of $CH_4$ emissions. The median, standard deviation, and the probability ($P_0$) of the correlated response to selection to be higher (lower) than 0 when the correlated response was positive (negative) were computed.

*Co-abundance network analysis of host genomic effects on the rumen microbiome.* To study the correlation structure among host genomic effects of the log-transformed abundances of 1107 microbial genera, 225 RUGs, and 1141 microbial genes, we built a co-abundance network analysis using deregressed host genomic effects (dGEBVs) for all microbial traits in the 359 samples. Deregressed host genomic effects were calculated from previously described univariate GBLUP models by using ACCF90 and DEPROOF90 programs[159]. Co-abundance network (Graphia software[164]) connected or edged microbial traits (nodes) based on a Pearson correlation >0.70 among their dGEBVs. The complexity of the graph was reduced by discarding nodes with a minimum number of incident edges (referred to as node degree) of 2, i.e. only those microbial traits Pearson-correlated (>0.70) with at least other 2 microbial traits were kept. The total number of microbial genera, RUGs, and microbial genes included in the network was 2129 out of the 2473 tested. The number of edges of each node was reduced by ranking the edges based on $k$-nearest neighbour algorithm and retaining only 80% of them. The software applies Markov Clustering algorithm by a flow simulation model[165] to find discrete groups of nodes (clusters) based on their position within the overall topology of the graph. The granularity of the clusters, i.e. the minimum number of nodes that a cluster has to contain, was set to 2 nodes. The network showed 106 clusters, but only those 12 clusters including ≥3 methanogenic archaea genera, RUGs and microbial genes involved in $CH_4$ metabolism pathway according to

KEGG[146] database or microbial genera/RUGs/genes host-genomically correlated with $CH_4$ emissions ($P_0 \geq 0.95$) were studied in depth.

*Enrichment analysis of microbial gene abundances in RUGs.* To identify which of the 225 RUGs were carrying the microbial genes (KO) demonstrating a $r_{gCH4}$ with a confidence level $P_0 \geq 0.95$, an enrichment analysis was performed by counting the number of unique proteins clustered in the 115 microbial genes mapped in each of the 225 RUGs.

*Identification of most informative microbial traits to predict $CH_4$ emission host genomic effects and maximize response to selection.* Only microbial variables present in the 359 animals, showing a RA ≥ 0.01%, with significant $h^2$ ($P \leq 2.02 \times 10^{-5}$), and host-genomically correlated with $CH_4$ emissions ($P_0 \geq 0.95$) were considered for breeding purposes. Four microbial genera and 36 microbial genes met these conditions. Due to computation reasons, only 30 microbial gene abundances were carried forward for downstream analysis. To use microbial gene information to select hosts emitting less $CH_4$, the estimation between their host genomic and residual (co)variance matrices was required. Host-genomic and residual (co)variances among the 30 selected microbial gene abundances were estimated using 435 bivariate analyses. Bivariate analyses fitted the same model as previously described for estimation of $r_{gCH4}$ with the same assumptions (Eqs. (8) and (9)). Results were based on Markov chain Monte Carlo chains consisting of 1,000,000 iterations, with a burn-in period of 200,000, and only 1 of every 100 samples was saved for inferences. Convergence was tested with POST-GIBBSF90 program by checking Z criterion of Geweke to be between −0.12 and 0.15. Monte Carlo sampling errors were computed using time-series procedures and checked to be at least 10 times lower than the standard deviation of the posterior marginal distribution[158]. The $31 \times 31$ host-genomic and residual variance–covariance matrices, including $CH_4$ emissions and the 30 microbial genes were built based on medians of the estimated variance components from the bivariate analyses and mean across all previous bivariate models for host genomic and residual variances of $CH_4$ emissions. Both matrices needed bending to be positive definite (tolerance for minimum eigenvalues = 0.001). The difference between original and bent matrices was never higher than the posterior standard error of the corresponding parameters.

*Estimation of the selection response of $CH_4$ emissions based on different sources of information.* We analysed three different scenarios to predict host-genomic effects of $CH_4$ emissions: (1) by using measured $CH_4$ emissions only, (2) by using the 30 microbial gene abundances only, and (3) by using a combination of both, measured $CH_4$ emissions and the 30 microbial gene abundances. The three scenarios were computed with data from 285 animals with $CH_4$ emissions and metagenomics information. All scenarios were calculated by GBLUP analysis assuming as fixed variance components the previously estimated $31 \times 31$ host genomic and residual variance-covariance matrices of the traits after bending. Scenario (1) was performed using a univariate GBLUP analysis including only measured $CH_4$ emissions; scenario (2) was computed by fitting a multivariate GBLUP model including the 30 microbial gene abundances host-genomically correlated to $CH_4$ emissions (using measured $CH_4$ emissions as missing value[166]); and scenario (3) considered besides the abundance of the 30 microbial genes, the measured $CH_4$ emission values. In all cases, models included the same effects as in Eq. (3). Host genomic values estimates for $CH_4$ emissions were based on Markov chain Monte Carlo chains consisting of 100,000 iterations, with a burn-in period of 20,000, and to reduce autocorrelation only 1 of every 100 samples was saved for inferences. Response to selection was estimated as the marginal posterior distributions of the difference between the mean of $CH_4$ emissions host genomic values of all animals with data and the mean of selected animals when alternatively, 1, 5, 10, 20, 30, 40, and 50% of our population were selected. The mean accuracy of the $CH_4$ emissions genomic values in each scenario was estimated as the average of the individual accuracies:

$$\text{Accuracy}_i = \sqrt{1 - \frac{\text{sd}_i^2}{g_{RMii}\sigma_{CH4}^2}}, \qquad (11)$$

where $\text{sd}_i$ is the standard deviation of the posterior marginal distribution of the host genomic value for animal $i$ and $g_{RMii}$ is the $G_{RM}$ diagonal element for animal $i$.

**Reporting summary.** Further information on research design is available in the Nature Research Reporting Summary linked to this article.

## Data availability
Metagenomic sequence reads for all rumen samples are available under European Nucleotide Archive (ENA) under accession projects PRJEB31266, PRJEB21624, and PRJEB10338. The genotypes of the host animals are readily available from the authors.

## Code availability
Metagenomic data processing was carried out using Kraken (https://ccb.jhu.edu/software/kraken/) for taxonomic annotation and Novoalign (http://www.novocraft.com/

support/download/ available under license) for functional annotation. SNP data filtering was performed PLINK (https://www.cog-genomics.org/plink2). Host genomic analysis were carried out using the RENUMF90, THRGIBBSF90, POSTGIBBSF90, ACCF90, and DEPROOF90 software, which have free access in http://nce.ads.uga.edu/wiki/doku.php?id=application_programs, except for ACCF90 and DEPROOF90 available only under research agreement. Network analysis was carried out by free access to Graphia software whose code source can be found at https://graphia.app/download.html.

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

## Acknowledgements

The authors thank Professor Ignacy Misztal and Dr. Shogo Tsuruta for making software available to us, Professor Agustín Blasco and Professor Chris Haley for their statistic advice, and Professor Michael Greenacre for his advice on compositional data analysis. We also thank Bin Zhao for his contribution to the identification and biological description of metagenomics data and Dr. Larissa Zetouni for her comments on the manuscript.

## Author contributions

M.M.-A., R.R., and M.W. conceived and designed the overall study, and M.M-A., M.W., and R.R. conceived, designed, and executed the bioinformatics analysis. M.D.A., C.-A.D., R.J.D., and M.A.C. provided essential insight into microbiology, rumen metabolism, nutrition, methane emissions and animal breeding. M.M.-A. and R.R. wrote the initial draft, and subsequently, all authors contributed intellectually to the interpretation and presentation of the results in the manuscript, which was edited and approved by all authors.

## Competing interests

The authors declare no competing interests.
