## [Peer Review File · Communications Biology]

Reviewers' comments:

Reviewer #1 (Remarks to the Author):

The manuscript quantifies and reports the effects of host genetics on the abundance of microbial genera and their genes characterized in relation to CH₄, by fitting GBLUP univariate animal models to estimate heritabilities for these abundance "traits". Moreover, the study reports estimates of selection response of CH₄ emissions by using information on microbial gene abundances with or without information on measured methane emission. The authors have undertaken several analysis and included in the manuscript a number of results to support their conclusion yet managed to keep a nice balance between the information included and the level of details. I upload them for that. The manuscript is also very well written and easy to read. Having said that, I have few concerns that I would appreciate if the authors can reflect on:

1) Assumption of similar variances across breeds

Major conclusions in the manuscript about "significant host genomic effects were drawn based on the heritability estimates fitting methane emission and microbial and their gene abundances as "traits"". Yet this central result is based on an analysis that unrealistically assumes homogenous variances across the different breeds of individuals making up the data. Breed is fitted as fixed effect, which might help account for differences in means but the variances could as well be different across these breeds, which is not accounted for by fitting the fixed effect of breed. Thus, heritability might be very different for the different breeds. I appreciate if the authors reflect on this.

2) Some clarity (lack of it) in numbers shown in the methods section

Line 598: "the data were obtained from 363 steers.."

Which data? Because there are different figures in the manuscript and there should be clarity as to what kind of data is available for what proportion of the animals. The heading in this paragraph is on the methane emission data. If the data is only obtained from 285 animals, as alluded to in line 605, this figure should be consistently mentioned in this paragraph. If the 383 figure is for the genotype data only, the figure must then be removed from the paragraph where the methane data is presented.

Line 624: "in total, 386 animals...". I don't get how the authors end up having genotypes for 386 animals after quality control on both SNPs and individuals when they initially (line 612) genotyped only 370 animals (363 steers + 7 sires)? One of the numbers is not right or this paragraph is not clear enough

Minor (editorial)

-line 19: "in" should be changed to "is"?

-line 58-59: Insert "been" after "...in the rumen have recently"

Reviewer #2 (Remarks to the Author):

The progression from traditional single ribosomal gene sequencing of microbes to full assemblies and gene abundances is a natural progression for the rumen and methane research. This group is world class for this kind of research. This type of work is timely on an important global topic. The authors have done a masterful job blending multidisciplinary research.

I am concerned about the study design and wish that this was not the case – because my other comments to the manuscript were minimal. I will be very relieved if clarification or additional steps can be taken to restore my faith in the design and estimates of heritability and genetic correlations.

A very large part of this study is based on genomic selection parameters and results. Genomic parameters such as heritability and genetic correlations are assuming within breed variation. This

study made use of multiple breeds for a rotational cross, this makes the population genetically diverse and heterogenous. When estimating genetic parameters such as heritability and genetic correlations these parameters are inflated when using genetically heterogenous populations. The bias of parameter estimates is further compounded by the numerically very small genomic reference population $n = (365)$ to add to this not all cows were phenotyped for CH₄ or DMI $n = (285)$ making the population even smaller. Genetic correlations estimated in such smaller number of animals is really to be regarded with high uncertainty. The authors also unfortunately used two very extreme diets contrasted for forage to concentrate ratio, these diets have the effect of inflating the variation in methane production and although the authors tried to control for this by dividing CH₄/DMI it cannot full account for this. When looking closer at supplementary 9 there is clear confounding between the breed types, diets and years. For example Aberdeen Angus cross is confounded with Forage diet in 2014 and 2017. Or Luining and Charlois are confounded with 2012 and 2013. When we then take a closer look at sequencing platforms, which are known to affect read depth we see strong confounding between sequencing platform and years lines 631 – 636. This is not accounted for in the models estimating heritability. I find that the poor design means I have little faith in the heritability estimates and genetic correlations upon which the manuscript is based.

Was there multiple testing correction conducted on the thousands of heritability estimates?

Using CH₄/DMI is a very peculiar choice. A breeding goal would need both DMI and CH₄ for be added to the selection index and appropriate weights given. How robust are the results when looking at CH₄ in g/day and DMI in kg/day directly?

Abstract

Lines 13 / 19 Two sentences both making a same statement about substantial genetic variation, suggest revising

Lines 27 / 28 consistency of brackets around %

Lines 32 / 35 / long run along sentence / suggest making two distinct sentences

Line 35 / 38 complex sentence suggest revising to make it easier to read

Line 39 /40. Multiple references have already concluded this <https://doi.org/10.3168/jds.2020-19889> <https://doi.org/10.1139/cjas-2019-0032>

Line 58 – 59 seems grammatically incorrect '...been identified'

Line 64 -66 Not entirely true – multiple references have suggested ways to approach this, you seem to have overlooked an entire field dedicated to microbiome driven selection strategies. see

Weishaar et al 2019 <https://doi.org/10.1111/jbg.12447>, Perez-Encisco et al 2021

<https://doi.org/10.1186/s12711-021-00658-7> , Saborio-Montero et al 2021

<https://doi.org/10.1016/j.livsci.2021.104538>.

Introduction – perhaps mention how your approach overcomes the ability of some microbial organisms to integrate foreign DNA from other microbial organisms into their DNA. This redundancy can only be captured by your approach and is not captured by the single gene sequencing approaches.

Line 108. What is 'large insight' ?

Line 110 – Cost Effective? I dispute this. Can you provide the cost of recording CH₄ per cow and the cost of obtaining your microbiome gene abundancies?

Methods:

Line 603-605. A reference to the power analysis or some explanation in supplementary materials in required.

Line 605. Starting a sentence with a number or abbreviation should be avoided where possible.

Line 611. 620

Line 611. Is CH₄/DMI the average CH₄ over 48 hr divided by the average DMI consumed over 48 hours?

Line 625 – 629. Provided the time difference between the CH₄ measurement and the post mortem rumen sampling is minimal and standardised.

Line 640 – what is RA?

Line 127. Title of this section reads 'Host genetics shapes the microbiome' is this really true? I think your results show that some of the variance in microbiome is heritable when controlling for other factors like diet and year see this article in humans where diet and housing were not standardised and the bold title <https://doi.org/10.1038/nature25973>. It seems that host and

environment share the microbiome and bold statements about what shapes the microbiome should be avoided.

Lines 206 – 365. This section starts with estimating genetic correlations between Genera RUGs and microbial gene abundance with CH₄ emission. The majority of this section naturally focuses on the genetic correlations between gene abundance and CH₄ emissions as this is the most novel. However the vast majority of genetic correlations between gene abundancies and CH₄ emission are negative. The authors go into details of how this could be due to inhibiting methanogenic substrates. However, the majority of genetic correlations between genera and RUG abundance are positive with CH₄ emission and opposite in sign. I feel that this discrepancy needs closer examination or clarification. How can the number of bacterial genera and RUGS increase CH₄ emission but the copy number of genes they are comprised of inhibit methanogenesis? What is more alarming here is that the genetic correlations with methane across genera, RUGs and gene abundance reported are changing sign and in the very extreme manner nearing the bounds of -1 and 1.

Line 441. Something wrong with the wording of this sentence.

Line 511- 514. This sentence claims that genomic selection on the functional microbiome associated with CH₄ microbiome-driven breeding will be of major importance. This is really unlikely, the authors are collecting rumen samples post mortem, it took many years to get a very small population size, which is far too small to be used in practice.

Lines 631 – 634 – three or four difference methods of sequencing – has to affect read number and quality – such matrix effects are not included in the models for heritability or genetic correlation estimation and almost certainly confounded with experimental year.

Lines 582 – 585. In practice a 30 trait model is unlikely to converge even with hundreds of thousands of animals. It is very challenging to see how you could advocate this selection strategy when it took substantial amount of time and resources to develop these 30 traits and a modest genomic reference population. I have asked above for clarification on the cost of phenotyping 1 animal for CH₄/DMI vs the suite of 30 gene abundances.

Line 759 – Are you certain your selection response strategy is correct? Particularly the intensity of selection. You are using a deceased genomic reference population, so the only feasible way to use this for breeding is to generate the EBVs on the Sire level.

Author's responses to comments of the Editor and Reviewers

Comment of the Editor:

Your manuscript entitled "Bovine host genome acts on specific metabolism, communication and genetic processes of rumen microbes host-genomically linked to methane emissions" has now been seen by 2 referees, whose comments are appended below. You will see from their comments that while they find your work of potential interest, they have raised quite substantial concerns. In light of these comments, we cannot accept the manuscript for publication. While our decision is to reject, we may be able to consider an appeal should the inclusion of additional data allow you to address the reviewer concerns regarding experimental design.

We hope you will find the referees' comments useful as you decide how to proceed. Should further experimental data or analysis allow you to address these criticisms, we would be happy to consider an appeal. However, please bear in mind that we will be reluctant to approach the referees again in the absence of major revisions. In particular, please note that the following revisions would be necessary for us to reconsider the manuscript:

Both referees raise issues with the ability of the experimental design to actually infer heritability. If you consider re-submitting the manuscript, I would need you to clarify the experimental design issues and ensure that your results are robust despite the clear confounding at **nutritional, host genetic and microbial sequencing levels**. The other constructive comments by both reviewers should of course be taken into account, too.

Author's Response :

We thank the referees for their criticisms of our experimental design, which showed that we did not address this issue sufficient in our original manuscript. We have carried out detailed further analyses to show that the different breeds or diets in our population have been appropriately adjusted and do not interfere with the estimation of genetic parameters.

We applied an optimal adjustment of the fixed effects as combination of experiment, breed and diet so that adjustment of each of these effects and their interactions was achieved. In the revised version of the manuscript we are including an exploratory analysis of the effect of breed, diet and experiment on methane emissions (Supplementary Table 10) and microbial traits to show that their adjustment as a combined fixed effect in genomic models is appropriate (lines 737-758). Using this fixed effect model, the impact of the used sequencer is indirectly adjusted for by using the effect of experiment in the model in which the sequencer effect is nested. We are not interested in estimation of these effects and their interactions; we are only interested in elimination of these nuisance effects so that the estimation of genomic parameters is unaffected by these effects. In addition, it has to be considered that each experiment was balanced for breed and diet effects as well as even the progenies of each sire were balanced over each diet so that the experimental design by itself has high power to disentangle genomic effects from diet effects (comment included in lines 635-637). To further analyse whether the estimation of genomic parameters is unaffected by these effects we found that the genomic variances of methane emissions as well as of the abundances of the main microbial traits were homogeneous between breeds and diets (lines 461-486). This last analysis is based on the posterior distributions of the EBVs computed and compared separately for each of the 4 breeds, and each of the 2 diets (as proposed by Sorensen et al. 2001). We obtained homogeneous genomic variances of all breeds or diets for methane emissions and microbial traits (Supplementary Figures 4 and 5).

Sorensen, D., Fernando, R. & Gianola, D. Inferring the trajectory of genetic variance in the course of artificial selection. *Genet. Res.* 77, 83–94 (2001).

Comments of Reviewer#1:

The manuscript quantifies and reports the effects of host genetics on the abundance of microbial genera and their genes characterized in relation to CH₄, by fitting GBLUP univariate animal models to estimate heritabilities for these abundance "traits". Moreover, the study reports estimates of selection response of CH₄ emissions by using information on microbial gene abundances with or without information on measured methane emission. The authors have undertaken several analysis and included in the manuscript a number of results to support their conclusion yet managed to keep a nice balance between the information included and the level of details. I upload them for that. The manuscript is also very well written and easy to read. Having said that, i have few concerns that i would appreciate if the authors can reflect on:

1) Assumption of similar variances across breeds. Major conclusions in the manuscript about "significant host genomic effects where drown based on the heritability estimates fitting methane emission and microbial and their gene abundances as "traits"". Yet this central result is based on an analysis that unrealistically assumes homogenous variances across the different breeds of individuals making up the data. Breed is fitted as fixed effect, which might help account for differences in means but the variances could as well be different across these breeds, which is not accounted for by fitting the fixed effect of breed. Thus, heritability might be very different for the different breeds. I appreciate if the authors reflect on this.

Author's Response:

Thank you very much for your and reviewers 2 comment, which shows that we have not addressed the issue of different breeds and diets sufficiently in our earlier manuscript. Therefore, we have now included in the manuscript a section showing that we have achieved robust parameters after adjustment of these fixed effects as combination of experiment, breed and diet (lines 737-758) and that the genomic variances are not different between breeds or diets (lines 461-486). The new section added to the revised manuscript is as follows:

Influence of breed, diet and experiment on CH₄ emissions and microbial traits. In our genomic models, we applied an optimal adjustment of the fixed effects as combination of experiment, breed and diet so that adjustment of each of these effects and their interactions was achieved (see below section: Estimation of host genomic parameters of CH₄ emissions and microbial traits). Sequencer effects were except for only 4 samples nested within experiment and therefore were accounted for due to inclusion of this effect. Since these are nuisance effects with a potential of influencing the estimation of genomic effects, we carried out further exploratory analyses and revealed that they were appropriately adjusted and do not interfere with the estimation of genomic parameters. In this exploratory analyses, we evaluated the effect of breed, diet, and experiment on microbial genes, genera and RUGs using a PERMANOVA analysis with 999 permutations computed with the R package *vegan*¹⁵⁶. Diet and experiment showed the largest effects on the composition of the rumen microbiome. Diet accounted for 18.9%, 12.0% and 13.19% of the total phenotypic variance in the microbial genes/microbial genera/RUGs databases, whilst experiment explained 7.5%, 11.17% and 6.09%, respectively (*P-value* < 0.001 in all cases). Breed effects on microbial genes/microbial genera/RUGs databases were depreciable (0.49%, 0.65% and 0.78%) and non-significant (*P-values* = 0.668, 0.385 and 0.141) indicating that the breeds considered in this study did not

show substantial differences in their functional or taxonomical microbiome composition. As expected, all effects became negligible (explaining 0% of the phenotypic variance) after their adjustment as a combined fixed effect. We also evaluated the effects of breed and diet on CH₄ emissions mean and variance (Supplementary Table 10a, b). Before adjustment, diet had a significant (P -value = 2.2×10^{-16}) a large effect on CH₄ (explained 41.2% of the phenotypic variance), whilst the effect of breed (2.92%, P -value = 0.0007) was smaller but still significant, however, both turned to 0% after the fixed-effects adjustment. Importantly, for the reliability of estimation of genetic parameters, animals from different breeds, or fed different diets presented homogenous CH₄ phenotypic variances (P -value of Levene test = 0.19 and 0.48).

Robustness of estimation of genomic parameters of CH₄ emissions and microbiome traits from a cross-classified design of breeds and basal diets. The data, from the highly environmentally standardised experiments, comprised of animals from different breeds that were offered different diets, which could be challenging as different breeds might have different genomic backgrounds for the analysed traits, and different diets could have inflated their variances. To consider the difference in means of these effects we fitted a combination of experiment, breed and diet effects so that an adjustment of each of these effects and their interactions was achieved. To analyse whether after this adjustment the genomic variances of CH₄ emissions and the abundances of 10 microbial genes selected for microbiome-driven breeding are homogeneous across breeds and diets, we computed their posterior distributions separately (Supplementary Fig. 4 and 5), following the partition of the variance suggested by Sorensen *et al.*¹²⁶, and recently used by Lara *et al.*¹²⁷. Using this methodology, we found that our genomic parameter estimates for CH₄ emissions are based on similar genomic variances across breeds (with medians of 3.8, 3.7, 4.0 and 3.9 (g/kg DMI)² for Aberdeen Angus, Limousin, Charolais crosses and pure breed Luining, respectively) and diets (with medians of 3.8 and 3.9 (g/kg DMI)² for Forage and Concentrate based diets) with almost entirely overlapping distributions indicating their homogeneity (Supplementary Fig. 4). We also identified homogeneous genomic variances for the abundances of microbial genes with overlapping distributions across breeds (Supplementary Fig. 5a) and diets (Supplementary Fig. 5b); for example, we estimated genomic variances with medians between 0.024-0.028 (g/kg DMI)² across breeds and 0.028-0.029 (g/kg DMI)² across diets for *RP-L35*, or between 0.25-0.30 (g/kg DMI)² across breeds and 0.292-0.296 (g/kg DMI)² across diets for *cofG* (Supplementary Figure 5). These results indicate that the data recorded under controlled experimental conditions with a cross-classified breed and diet two-way experimental design and progeny groups balanced over diets resulted in reliable and robust genomic parameters estimates.

In summary, our results show:

Influence of breed, diet and experiment on CH₄ emissions and microbial traits. We estimated the effect of breed and diet on CH₄ at phenotypic level (Table S10a, b) and showed that breed and diet are successfully adjusted for in the model. We also carried out a Levene test to show that CH₄ shows homogenous phenotypic variances across breeds and diets. This is also noticeable in Supplementary Figure S1. We also estimated the effect of breed, diet and experiment (nested with sequencer effect) on the microbiome composition at phenotypic level and demonstrated that they are successfully adjusted in the model and do not have a significant effect after such adjustment.

Genomic variance partition across breeds and diets. To study the homogeneity of genomic variances for CH₄ emissions and microbial traits within each breed and within each diet, we

computed the marginal posterior distribution of the genomic variance for methane emissions and important microbial traits selected for microbiome-driven breeding in each of the 4 different breeds and in each of the 2 different diets following the same basis as suggested in Sorensen *et al.*¹, and more recently, in Lara *et al.*² We show that our model captures similar genomic variances across breeds for CH₄ (Figure S4) and microbial genes (Figure S5).

Sorensen, D., Fernando, R. & Gianola, D. Inferring the trajectory of genetic variance in the course of artificial selection. *Genet. Res.* 77, 83–94 (2001).

de C. Lara, L. A., Pocrnic, I., Gaynor, R. C. & Gorjanc, G. Temporal and genomic analysis of additive genetic variance in breeding programmes. *bioRxiv* 2020.08.29.273250 (2020).

2) Some clarity (lack of it) in numbers shown in the methods section Line 598: “the data were obtained from 363 steers..” Which data? Because there are different figures in the manuscript and there should be clarity as to what kind of data is available for what proportion of the animals. The heading in this paragraph is on the methane emission data. If the data is only obtained from 285 animals, as alluded to in line 605, this figure should be consistently mentioned in this paragraph. If the 383 figure is for the genotype data only, the figure must then be removed from the paragraph where the methane data is presented.

Author’s Response:

Thank you for asking to provide further clarification. The text has been amended to improve its clarity.

Line 624: “in total, 386 animals...”. I don’t get how the authors end up having genotypes for 386 animals after quality control on both SNPs and individuals when they initially (line 612) genotyped only 370 animals (363 steers + 7 sires)? One of the numbers is not right or this paragraph is not clear enough.

Author’s Response:

Thanks for your comment. We explained in the revised manuscript that, additionally to the 363 steers, we had 23 semen samples + 7 blood samples available from sires of the steers (n=393 in total). Of those, 7 did not pass the quality control of call rate higher equal 90% so that 386 remained for the analysis. We have revised the section of ‘Host genomic samples’ based on the reviewer’s comments.

Minor (editorial)

-line 19: “in” should be changed to “is”? **Changed as suggested by the reviewer.**

-line 58-59: Insert “been” after “..in the rumen have recently” **Changed as suggested by the reviewer.**

Comments of Reviewer #2:

The progression from traditional single ribosomal gene sequencing of microbes to full assemblies and gene abundances is a natural progression for the rumen and methane

research. This group is world class for this kind of research. This type of work is timely on an important global topic. The authors have done a masterful job blending multidisciplinary research. I am concerned about the study design and wish that this was not the case – because my other comments to the manuscript were minimal. I will be very relieved if clarification or additional steps can be taken to restore my faith in the design and estimates of heritability and genetic correlations. A very large part of this study is based on genomic selection parameters and results. Genomic parameters such as heritability and genetic correlations are assuming within breed variation. This study made use of multiple breeds for a rotational cross, this makes the population genetically diverse and heterogeneous. When estimating genetic parameters such as heritability and genetic correlations these parameters are inflated when using genetically heterogeneous populations. The bias of parameter estimates is further compounded by the numerically very small genomic reference population $n = (365)$ to add to this not all cows were phenotyped for CH₄ or DMI $n = (285)$ making the population even smaller. Genetic correlations estimated in such a smaller number of animals is really to be regarded with high uncertainty. The authors also unfortunately used two very extreme diets contrasted for forage to concentrate ratio, these diets have the effect of inflating the variation in methane production and although the authors tried to control for this by dividing CH₄/DMI it cannot fully account for this.

Author's Response:

As already indicated in the response to comments of Reviewer 1, we also thank you for this comment, which shows that we have not addressed the issue of different breeds and diets sufficiently in our earlier manuscript. Therefore, we have now included in the manuscript a section showing that we have achieved robust parameters after adjustment of these fixed effects as combination of experiment, breed and diet and that the genomic variances are not different between breeds or diets:

Influence of breed, diet and experiment on CH₄ emissions and microbial traits. In our genomic models, we applied an optimal adjustment of the fixed effects as combination of experiment, breed and diet so that adjustment of each of these effects and their interactions was achieved (see below section: Estimation of host genomic parameters of CH₄ emissions and microbial traits). Sequencer effects were except for only 4 samples nested within experiment and therefore were accounted for due to inclusion of this effect. Since these are nuisance effects with a potential of influencing the estimation of genomic effects, we carried out further exploratory analyses and revealed that they were appropriately adjusted and do not interfere with the estimation of genomic parameters. In this exploratory analyses, we evaluated the effect of breed, diet, and experiment on microbial genes, genera and RUGs using a PERMANOVA analysis with 999 permutations computed with the R package *vegan*¹⁵⁶. Diet and experiment showed the largest effects on the composition of the rumen microbiome. Diet accounted for 18.9%, 12.0% and 13.19% of the total phenotypic variance in the microbial genes/microbial genera/RUGs databases, whilst experiment explained 7.5%, 11.17% and 6.09%, respectively (P -value < 0.001 in all cases). Breed effects on microbial genes/microbial genera/RUGs databases were depreciable (0.49%, 0.65% and 0.78%) and non-significant (P -values = 0.668, 0.385 and 0.141) indicating that the breeds considered in this study did not show substantial differences in their functional or taxonomical microbiome composition. As expected, all effects became negligible (explaining 0% of the phenotypic variance) after their adjustment as a combined fixed effect. We also evaluated the effects of breed and diet on CH₄ emissions mean and variance (Supplementary Table 10a, b). Before adjustment, diet had a significant (P -value = 2.2×10^{-16}) large effect on CH₄ (explained 41.2% of the phenotypic variance), whilst the effect of breed (2.92%, P -value = 0.0007) was smaller but still significant,

however, both turned to 0% after the fixed-effects adjustment. Importantly, for the reliability of estimation of genetic parameters, animals from different breeds, or fed different diets presented homogenous CH₄ phenotypic variances (*P-value* of Levene test = 0.19 and 0.48).

Robustness of estimation of genomic parameters of CH₄ emissions and microbiome traits from a cross-classified design of breeds and basal diets. The data, from the highly environmentally standardised experiments, comprised of animals from different breeds that were offered different diets, which could be challenging as different breeds might have different genomic backgrounds for the analysed traits, and different diets could have inflated their variances. To consider the difference in means of these effects we fitted a combination of experiment, breed and diet effects so that an adjustment of each of these effects and their interactions was achieved. To analyse whether after this adjustment the genomic variances of CH₄ emissions and the abundances of 10 microbial genes selected for microbiome-driven breeding are homogeneous across breeds and diets, we computed their posterior distributions separately (Supplementary Fig. 4 and 5), following the partition of the variance suggested by Sorensen *et al.*¹²⁶, and recently used by Lara *et al.*¹²⁷. Using this methodology, we found that our genomic parameter estimates for CH₄ emissions are based on similar genomic variances across breeds (with medians of 3.8, 3.7, 4.0 and 3.9 (g/kg DMI)² for Aberdeen Angus, Limousin, Charolais crosses and pure breed Luig, respectively) and diets (with medians of 3.8 and 3.9 (g/kg DMI)² for Forage and Concentrate based diets) with almost entirely overlapping distributions indicating their homogeneity (Supplementary Fig. 4). We also identified homogeneous genomic variances for the abundances of microbial genes with overlapping distributions across breeds (Supplementary Fig. 5a) and diets (Supplementary Fig. 5b); for example, we estimated genomic variances with medians between 0.024-0.028 (g/kg DMI)² across breeds and 0.028-0.029 (g/kg DMI)² across diets for *RP-L35*, or between 0.25-0.30 (g/kg DMI)² across breeds and 0.292-0.296 (g/kg DMI)² across diets for *cofG* (Supplementary Figure 5). These results indicate that the data recorded under controlled experimental conditions with a cross-classified breed and diet two-way experimental design and progeny groups balanced over diets resulted in reliable and robust genomic parameters estimates.

When looking closer at supplementary 9 there is clear confounding between the breed types, diets and years. For example, Aberdeen Angus cross is confounded with Forage diet in 2014 and 2017. Or Luig and Charlois are confounded with 2012 and 2013. When we then take a closer look at sequencing platforms, which are known to affect read depth we see strong confounding between sequencing platform and years lines 631 – 636. This is not accounted for in the models estimating heritability. I find that the poor design means I have little faith in the heritability estimates and genetic correlations upon which the manuscript is based.

Author's Response:

We have considered the data distribution across breeds, diets and experiments in the statistical model as combination of experiment, breed and diet so that adjustment of each of these effects and their interactions was achieved. The sequencing platform has been also taken into account by this adjustment as it is nested within experiment, as explained in methods (lines 740-741). We want to emphasis again, that the data were obtained under control experimental conditions, with measurement of methane using the gold standard of respiration chambers, using microbial genera, RUG and genes based on deep whole genome

sequencing, using a high-density SNP panel of the host genome, with fixed effects of breed and diet cross-classified and sire progeny groups balanced over diets. This means it is a unique data using the appropriate statistical model which resulted in highly reliable and unique genetic parameters. Our further analyses of the robustness of the genetic parameter estimates presented above provide evidence that these estimates are reliable. In addition we have considered stringent multiple testing corrections to identify significant genetic parameters.

Was there multiple testing correction conducted on the thousands of heritability estimates?

Author's Response:

Yes, we did account for multiple testing on heritability's estimates. The procedure is described in methods: "To test the significance of host genomic effects we analysed the fitness of the full univariate genomic model vs. the univariate model without host genomic effect by comparing the Deviance information criterion³ between models and computing and computing the Bayes Factor using an approximation of the marginal likelihood probability⁴. Bayes Factor was corrected for multi-testing by assuming prior odds ratios equal to 1/number of hypothesis tests performed (n=2,473). We assumed an evidence of a host genomic effect on the microbial trait when the DIC of the full model was at least 20 points lower than the DIC in the reduced model, and Bayes Factor was higher than 3⁵."

Using CH4/DMI is a very peculiar choice. A breeding goal would need both DMI and CH4 for be added to the selection index and appropriate weights given. How robust are the results when looking at CH4 in g/day and DMI in kg/day directly?

Author's Response:

We have now work in progress with regards to DMI (kg/day) with the potential to analyse its associations with CH4(g/day). However, this is beyond the scope of the present paper because an entire new trait of DMI has to be analysed with all microbial genera/RUGs and genes

Abstract

Lines 13 / 19 Two sentences both making a same statement about substantial genetic variation, suggest revising. Changed as suggested by the reviewer.

Lines 27 / 28 consistency of brackets around %. In line 27, 13% is in brackets to reflect complementary information to the statement.

Lines 32 / 35 / long run along sentence / suggest making two distinct sentences Changed as suggested by the reviewer.

Line 35 / 38 complex sentence suggest revising to make it easier to read Changed as suggested by the reviewer.

Line 39 /40. Multiple references have already concluded this <https://doi.org/10.3168/jds.2020-19889> <https://doi.org/10.1139/cjas-2019-0032> Changed as suggested by the reviewer and references added to the manuscript.

Line 58 – 59 seems grammatically incorrect '....been identified' Changed as suggested by the reviewer.

Line 64 -66 Not entirely true – multiple references have suggested ways to approach this, you seem to have overlooked an entire field dedicated to microbiome driven selection strategies. see Weishaar et al 2019 <https://doi.org/10.1111/jbg.12447>, Perez-Encisco et al 2021 <https://doi.org/10.1186/s12711-021-00658-7>, Saborio-Montero et al 2021 <https://doi.org/10.1016/j.livsci.2021.104538>.

Author's Response:

Our aim here is to highlight that a microbiome-driven breeding strategy based on the microbiome function, i.e. microbial genes, has been never explored before. We have clarified the sentence and included the work of Weishaar et al. 2019, Pérez-Enciso et al 2021 and Saborio-Montero et al., 2020 as references for microbiome-driven breeding based on the taxonomical composition of the microbiome (lines 65-66).

Introduction – perhaps mention how your approach overcomes the ability of some microbial organisms to integrate foreign DNA from other microbial organisms into their DNA. This redundancy can only be captured by your approach and is not captured by the single gene sequencing approaches.

Author's Response:

Thank you very much for this very interesting thought, which we have included into the introduction (lines 56-57).

Line 108. What is 'large insight' ? We have rewritten it as 'better understanding'

Line 110 – Cost Effective? I dispute this. Can you provide the cost of recording CH4 per cow and the cost of obtaining your microbiome gene abundancies?

Author's Response:

The cost for measuring animals in respiration chambers is about 1,500 £, whereby we have found an approach to get microbiome gene abundances including DNA extraction of rumen samples below £50, which can be even substantially reduced depending on the scale of use of this methodology. In addition, sequencing technology is advancing so rapidly that the costs of sequencing will decline substantially.

Methods:

Line 603-605. A reference to the power analysis or some explanation in supplementary materials is required.

Author's Response:

Additional explanations have been added to Methods section (lines 635-640 of the revised manuscript). The Power-Analysis was based on the obtained sire estimates (68 progeny in total) and the root MSE of 3.27 using the model including the effects of sire within breed type, diet, respiration chamber and randomised block as published in Roehe et al. (2016) and resulted in an achieved power of 0.93 to identify differences in sire effects on methane yield of their progeny in this first experiment. The high power was achieved with an average number of progenies per sire around 8 with 9 sires resulting in a total number of 72 cattle starting this experiment. Because 4 animals could not be recorded due to health issues or an air leak in one of the respiration chambers, only 68 animals were considered in the power analysis. Since all following up experiments had similar experimental designs, each experiment has the power to identify potential sire difference in methane yield which increases the power to estimate genetic parameters and is expected together with the standardisation of all other environmental effects (all animals were tested on the same farm under high standardised

management conditions) to be the main reasons that we obtained significant parameters after consideration of multiple testing in a relatively small population.

Line 605. Starting a sentence with a number or abbreviation should be avoided where possible.

Line 611. 620 Changed as suggested by the reviewer.

Line 611. Is CH₄/DMI the average CH₄ over 48 hr divided by the average DMI consumed over 48 hours? Yes, it has been clarified in the revised manuscript.

Line 625 – 629. Provided the time difference between the CH₄ measurement and the post mortem rumen sampling is minimal and standardised.

Author's Response:

The time difference between the CH₄ measurement and post-mortem rumen sampling was based on the logistics up to 3 weeks after leaving the respiration chambers but our previous (Snelling et al. and also on-going research shows that the rumen microbiome is very stable so that the time of taking the rumen samples is of low impact on the predictability of methane emissions.

Line 640 – what is RA? We have revised the manuscript to clearly show that RA means relative abundance.

Line 127. Title of this section reads 'Host genetics shapes the microbiome' is this really true? I think your results show that some of the variance in microbiome is heritable when controlling for other factors like diet and year see this article in humans where diet and housing were not standardised and the bold title <https://doi.org/10.1038/nature25973>. It seems that host and environment share the microbiome and bold statements about what shapes the microbiome should be avoided.

Author's Response:

Based on the reviewers comment we have modified the section title as "Host genomics affects the microbiome composition"

Lines 206 – 365. This section starts with estimating genetic correlations between Genera RUGs and microbial gene abundance with CH₄ emission. The majority of this section naturally focuses on the genetic correlations between gene abundance and CH₄ emissions as this is the most novel. However the vast majority of genetic correlations between gene abundancies and CH₄ emission are negative. The authors go into details of how this could be due to inhibiting methanogenic substrates. However, the majority of genetic correlations between genera and RUG abundance are positive with CH₄ emission and opposite in sign. I feel that this discrepancy needs closer examination or clarification. How can the number of bacterial genera and RUGS increase CH₄ emission but the copy number of genes they are comprised of inhibit methanogenesis? What is more alarming here is that the genetic correlations with methane across genera, RUGs and gene abundance reported are changing sign and in the very extreme manner nearing the bounds of -1 and 1.

Author's Response:

During writing of this research manuscript, we realised the same point as raised by the reviewer, and therefore included an enrichment analysis of the 115 microbial genes over the RUGs identified in our population (see Figure 3 and Supplementary Figure 3), aiming to match the genomic informative microbial genes across taxa. The RUGs most highly enriched with the 115 microbial genes in our population are not those with positive rgCH₄ reported in Table S3B (see Supplementary Table 6). Instead, many of these highly-enriched 20 RUGs are classified as uncultured Lachnospiraceae and showed negative rgCH₄ < -0.65 (P₀ ≥ 0.85), consistently with the microbial genes that they carry. In Tables S3A and S3B we also report specific microbial genera (Dorea and Balutia) and RUGs (RUG13438) classified within the Lachnospiraceae family with negative rgCH₄ and P₀ > 0.95. We also found that part of the 115 microbial genes occur in RUGs annotated as Eubacterium ruminatum, Eubacterium pyruvatorans or Kandleria vitulina, (Supplementary Figure 3), and both Kandleria and Eubacterium microbial genera show negative rgCH₄ when identified by alignment of the sequences to Hungate + Refseq databases) (Table S3A). We also matched our 115 microbial genes into 5 RUGs classified as uncultured Methanobrevibacter strains, some of them with negative rgCH₄, proposing the hypothesis of functional versatility amongst different Methanobrevibacter strains on CH₄ emissions. In conclusion, our results based on genera, RUG and microbial genes are consistent. We added information to the paragraph 'RUGs enriched with CH₄-related microbial genes are strongly host-genomically correlated to CH₄ emissions' to make this clear in the manuscript.

Line 441. Something wrong with the wording of this sentence. Thank you very much, we have revised the sentence.

Line 511- 514. This sentence claims that genomic selection on the functional microbiome associated with CH₄ microbiome-driven breeding will be of major importance. This is really unlikely, the authors are collecting rumen samples post mortem, it took many years to get a very small population size, which is far too small to be used in practice.

Author's Response:

Thank you very much for the comment. Yes, you are right that it took us very long time to obtain the unique data (methane emissions using respiration chambers in a population of designed genetic structure) of this study. However, this was a research project to understand the fundamentals of host genomics and microbiome composition. Based on this research, we have presently a project with the breeding company Genus where we are taking rumen samples and obtaining microbial gene abundance and are in the process to implement our research into practical breeding.

Lines 631 – 634 – three or four difference methods of sequencing – has to affect read number and quality – such matrix effects are not included in the models for heritability or genetic correlation estimation and almost certainly confounded with experimental year.

Author's Response:

We have included in our analysis the experiment-breed-diet effects into the statistical model so that the effect of the experiment, in which the sequence platform is nested, is accounting for the different methods of sequencing. In addition, we considering due to using the combined fixed effects of experiment x breed x diet all potential interactions among these effects.

Lines 582 – 585. In practice a 30 trait model is unlikely to converge even with hundreds of thousands of animals. It is very challenging to see how you could advocate this selection strategy when it took substantial amount of time and resources to develop these 30 traits and a modest genomic reference population. I have asked above for clarification on the cost of phenotyping 1 animal for CH4/DMI vs the suite of 30 gene abundances.

Author's Response:

We have provided fixed variance components (that we previously estimated) in the 30-trait model in order to estimate the host-genomic effects (as explained in the Method section), and we have checked the convergence of the EBVs chains. As an example, see below the MCMC chain diagnosis of the marginal posterior distributions of CH4 EBVs of 4 animals estimated under scenario 2 (i.e., exclusively using the information of the 30 microbial genes):

Density Plot:

Geweke Plot:

The present research manuscript is using the best information available in the data, which we identified would be in this case a 30-trait model. However, in practice fewer traits could be used, canonical transformation could be implemented, etc. This means, there are many options possible coping with these practical challenges that are beyond the scope of this research manuscript. We want to emphasize that our main aim was to improve the understanding of the host genomics effects on the abundance of microbial genes to get a better understanding of the microbial functions influenced by host genomics.

Line 759 – Are you certain your selection response strategy is correct? Particularly the intensity of selection. You are using a deceased genomic reference population, so the only feasible way to use this for breeding is to generate the EBVs on the Sire level.

Author's Response:

We have carried out the selection on the given population. In practice it will be a much larger population with potentially much higher genetic variation and therefore higher expected response. Therefore, we have carried out the selection in the easiest way to just reflect the potential of breeding for methane emissions using the most obvious way given the population. In particular, we were interested in the potential of microbiome-driven breeding in comparison of direct breeding for measured methane emissions giving our population. In practical breeding there will be many different strategies of selection, which is way beyond the scope of this manuscript where we emphasize on the understanding of the host genomics on functional microbial genes.

REVIEWERS' COMMENTS:

Reviewer #1 (Remarks to the Author):

I thank the authors for addressing most of my comments/concerns.